# Enforcement of developmental lineage specificity by transcription factor Oct1

**Zuolian Shen[1†], Jinsuk Kang[1†‡], Arvind Shakya[1†§], Marcin Tabaka[2], Elke A Jarboe[1], Aviv Regev[2,3], Dean Tantin[1*]**

[1]Department of Pathology, University of Utah School of Medicine, Salt Lake City, United States; [2]The Broad Institute of MIT and Harvard, Cambridge, United States; [3]Howard Hughes Medical Institute, Massachusetts Institute of Technology, Cambridge, United States

*For correspondence: dean.tantin@path.utah.edu

†These authors contributed equally to this work

Present address: ‡C and C Research Laboratories, Suwon, South Korea; §Celgene Corporation, San Diego, United States

**Abstract** Embryonic stem cells co-express Oct4 and Oct1, a related protein with similar DNA-binding specificity. To study the role of Oct1 in ESC pluripotency and transcriptional control, we constructed germline and inducible-conditional Oct1-deficient ESC lines. ESCs lacking Oct1 show normal appearance, self-renewal and growth but manifest defects upon differentiation. They fail to form beating cardiomyocytes, generate neurons poorly, form small, poorly differentiated teratomas, and cannot generate chimeric mice. Upon RA-mediated differentiation, Oct1-deficient cells induce lineage-appropriate developmentally poised genes poorly while lineage-inappropriate genes, including extra-embryonic genes, are aberrantly expressed. In ESCs, Oct1 co-occupies a specific set of targets with Oct4, but does not occupy differentially expressed developmental targets. Instead, Oct1 occupies these targets as cells differentiate and Oct4 declines. These results identify a dynamic interplay between Oct1 and Oct4, in particular during the critical window immediately after loss of pluripotency when cells make the earliest developmental fate decisions.

## Introduction

The mammalian blastocyst inner cell mass (ICM) contains undifferentiated, pluripotent cells capable of generating all tissue lineages of the embryo proper. Cultured embryonic stem cells (ESCs) are derived from these cells and have similar capabilities (*Abranches et al., 2009*). The POU transcription factor Oct4/Pou5f1 is an indispensable component of the regulatory circuitry underlying these properties (*Morey et al., 2015*). It is expressed in the ICM and in ESCs where its loss accompanies differentiation (*Nichols et al., 1998*). Oct4 is also widely used to generate induced pluripotent stem cells (iPSCs) from somatic cells (*Takahashi and Yamanaka, 2006*).

Together with other factors, Oct4 sustains pluripotency by activating 'core' targets such as *Pou5f1* (encoding Oct4 itself) and *Nanog* (*Boyer et al., 2005*). It also maintains 'poised' targets, including developmentally critical transcription regulators, in a silent but readily inducible state (*Bernstein et al., 2006*; *Meissner et al., 2008*). These genes frequently encode developmentally important transcription factors and are marked with a bivalent chromatin signature defined by the simultaneous presence of H3K4me3 and H3K27me3 (*Azuara et al., 2006*; *Bernstein et al., 2006*; *Ku et al., 2008*; *Pan et al., 2007*).

Oct1/Pou2f1 is a widely expressed protein related to Oct4. The two proteins have similar DNA-binding specificity (*Tantin, 2013*). In somatic cells, it regulates stem cell and immune memory phenotypes (*Maddox et al., 2012*; *Shakya et al., 2015b*) and is associated with cytotoxic stress resistance, glycolytic metabolism and malignant transformation (*Bellance et al., 2012*; *Shakya et al., 2009*; *Tantin et al., 2005*). Oct1 amplification and/or overexpression correlates with tumor aggressiveness in esophageal, gastric, prostate, lung, cervical, and colorectal cancer (*Vázquez-*

**eLife digest** Humans and most other animals are composed of hundreds of different types of cell, including nerve cells, muscle cells and blood cells. Despite performing many different roles, these cells all develop from a single fertilized egg, which divides to make a particular group of cells that when studied in the laboratory are called embryonic stem cells (or ESCs for short).

The ability of a cell to become a different cell type is defined as "potency". ESCs are unique because they can specialize into any type of cell present in the adult organism, and they are therefore called "pluripotent". However, as the embryo develops, its ESCs gradually lose their potency, and become more and more specialized. The activity of a great number of genes must be regulated during the transition from pluripotent to specialized cells, and some of the mechanisms involved in this transition are still unclear.

ESCs are known to need a gene-regulating protein called Oct4 to remain pluripotent and Shen, Kang, Shakya et al. now show that a similar protein named Oct1 is essential for their transition to becoming more specialized. When the gene for Oct1 was deleted from mouse ECSs, they behaved largely like "normal" ESCs, but could not properly mature into certain cell types such as heart and nerve cells. Molecular analyses revealed that Oct4 and Oct1 compete to regulate the activity of many common genes with opposing outcomes: Oct4 keeps ESCs pluripotent while Oct1 leads them to specialize. The Oct4 protein is abundant in ESCs and prevails over Oct1, but as the cells mature, the levels of Oct4 drop, and Oct1 takes over in the regulation of their common target genes.

Going forward, a better understanding of how ESCs become specialized will help basic research in the laboratory and allow scientists to tackle new questions about how the human body develops and how our organs work. In the longer-term, these findings might also have applications in the field of regenerative medicine, which aims to repair or replace a person's cells, tissues or organs to improve their health.

*Arreguín and Tantin, 2016*). It is also co-expressed with Oct4 in ESCs (*Okamoto et al., 1990*; *Rosner et al., 1990*). Oct1-deficient mice undergo implantation but show defects following gastrulation, most prominently in extra-embryonic tissues, where trophoblast stem cell development is arrested and expression of the direct Oct1 target *Cdx2* is defective (*Sebastiano et al., 2010*). Tetraploid complementation bypasses this developmental restriction, allowing embryos to survive to E8.5–9.5 where they die from an embryo-intrinsic block. These embryos are runted, developmentally arrested, and lack beating hearts. (*Sebastiano et al., 2010*). A slightly less severe germline allele dies in mid-gestation and manifests runting, anemia, hemorrhaging, and other defects with variable penetrance (*Wang et al., 2004*).

Here, we show that ESCs lacking Oct1 have no discernable defects when maintained in an undifferentiated state, but that silent, normally poised developmental-specific genes fail to induce properly upon differentiation. Additionally, genes specific for alternative developmental lineages are inappropriately expressed. Most prominently, placenta-specific genes not normally expressed in any ESC-derived lineage are induced, indicating that Oct1 restricts extra-embryonic gene expression in differentiating ESCs. Additionally, these cells show phenotypic defects when differentiated into multiple lineages, form smaller and less differentiated teratomas, and fail to generate chimerism when injected into blastocysts. ChIPseq identifies a group of targets co-bound by Oct1 and Oct4 in ESCs associated with non-classical binding sites termed MOREs (More Palindromic Octamer Related Elements, ATGCATATGCAT). These sites are inducibly bound by Oct1 in somatic cells lacking Oct4. The function of Oct1 at these genes is to insulate their expression against repression by oxidative stress, and consistently Oct1-deficient ESCs are hypersensitive to oxidative stress. Oct1 associates with developmentally poised targets upon differentiation and Oct4 loss, explaining the altered gene expression observed with RNAseq. These results establish Oct1 as a key mediator of both developmental-specific gene induction and repression, and identify a dynamic interplay in which Oct1 replaces Oct4 at target genes as ESCs differentiate and early decisions about induction or repression of lineage-specific genes are made.

## Results

### Oct1 germline-deficient ESCs are phenotypically normal but differentiate abnormally

We derived Oct1-deficient ESC lines by intercrossing *Pou2f1* germline heterozygotes (*Wang et al., 2004*). Oct1-deficient animals die in utero (*Sebastiano et al., 2010*; *Wang et al., 2004*), but survive long enough to derive ESCs. Two Oct1-deficient lines and two littermate WT controls were generated. All had normal karyotypes (not shown). Oct1-deficient ESCs proliferate at normal rates (not shown), are morphologically normal (*Figure 1A*) and can be propagated for a month in culture with no loss of ESC morphology (not shown). They express normal levels of Oct4, Sox2, and Nanog protein but no Oct1 (*Figure 1B*). In addition, cells express the pluripotency-associated *Pou5f1* (Oct4), *Sox2*, *Nanog,* and *Dppa4* mRNAs at normal levels (*Figure 1C*). *Ahcy*, a stress-inducible Oct1 target in which the function of Oct1 is to prevent stress-associated repression (*Kang et al., 2009*; *Shakya et al., 2011*), was also unaltered.

To study differentiation, we used early-passage Oct1-deficient and WT control ESCs to form embryoid bodies (EBs). Oct1-deficient ESCs were able to aggregate into EBs at d four with morphology similar to WT (*Figure 1D*). Similar results were obtained at days 1 and 2 (*Figure 1—figure supplement 1A*). During EB formation, *Pou5f1* and *Sox2* were down-modulated with similar kinetics in Oct1-deficient and WT cells, while *Pou2f1* (Oct1) remained undetectable (*Figure 1E*). *Sox17* (endoderm), *Brachyury* (*T*, definitive mesoderm), and *Fgf5* (definitive ectoderm) expression in Oct1-deficient EBs was grossly similar to WT at some (days 4, 9, or 14) timepoints (*Figure 1F*), consistent with findings that Oct1 is dispensable for gastrulation (*Sebastiano et al., 2010*; *Wang et al., 2004*). However, there were consistent defects in expression in the Oct1-deficient condition at day 14 for *Sox17* and day 4 for *T* and *Fgf5*. *Sox17*, *T* and *Fgf5* are known Oct4 targets (*Chen et al., 2008*). By day 5, Oct1-deficient EBs were somewhat smaller in appearance (*Figure 1—figure supplement 1B*). We therefore looked for further evidence of defects in induction kinetics in three other known silent but developmentally inducible Oct4 target genes: *Hoxa5*, *Hoxc6*, and *Gata2* (*Chen et al., 2008*). Each of these genes showed a similar pattern of defective induction in Oct1-deficient EBs relative to WT controls (*Figure 1G*).

To study gene induction using a more developmentally restricted system, we analyzed expression of known developmentally inducible Oct4 target genes during RA-mediated differentiation of WT and Oct1-deficient ESCs. RA treatment of ESCs ultimately results in a largely neuronal phenotype, but waves of gene expression, differentiation, proliferation, and cell death take place during the course of RA treatment (*Walker et al., 2007*). Upon differentiation, ESCs ±Oct1 lose their clustered, spherical, refractile morphology with similar kinetics (not shown). *Pou5f1* and *Sox2* were also lost with similar kinetics ±Oct1, while *Pou2f1* was not detectable in KO cells (*Figure 2A*). To study developmental gene expression, we tested *Hoxa5*, *Hoxc6*, *Cdx2*, and *Sox17*. These genes encode developmentally important transcription factors and are known Oct4 targets (*Chen et al., 2008*), but are silent in ESCs. Upon RA-mediated differentiation, lineage-appropriate (ectoderm) genes such as *Hoxa5* and *Hoxc6* (*Jiang et al., 2011*) normally 'resolve' their bivalent state by losing H3K27me3 and becoming induced, while lineage-inappropriate (e.g. endoderm) genes such as *Cdx2* and *Sox17* normally resolve by losing H3K4me3, gaining DNA methylation, and becoming stably silenced. Induction of *Hoxa5* and *Hoxc6* was robust following RA-mediated differentiation of WT cells, but defective in the Oct1 KO condition (*Figure 2B*). In contrast, *Cdx2* was ectopically activated upon RA-mediated differentiation of Oct1-deficient ESCs (*Figure 2B*). Similarly, the definitive endoderm-specific gene *Sox17* is not normally induced upon RA-mediated differentiation, but showed ectopic expression in the absence of Oct1 (*Figure 2B*). The ectopic *Sox17* expression observed with RA differentiation differed from the expression defects observed in Oct1-deficient EBs, which include endodermal lineages and which showed 100-fold stronger *Sox17* expression (*Figure 1F*). These results indicate that Oct1-deficient ESCs induce lineage-appropriate developmental genes poorly, while ectopically expressing lineage-inappropriate genes.

In order to determine whether forced Oct1 expression during differentiation was sufficient to correct defects in gene expression, we differentiated Oct1-deficient ESCs using RA and infected the cell during the differentiation timecourse with lentiviral vectors expressing Oct1 and a puromycin resistance cassette, or empty vectors containing the puromycin resistance cassette alone. We confirmed that cells transduced with this vector overexpressed Oct1 by immunoblotting (*Figure 2C*). Cells

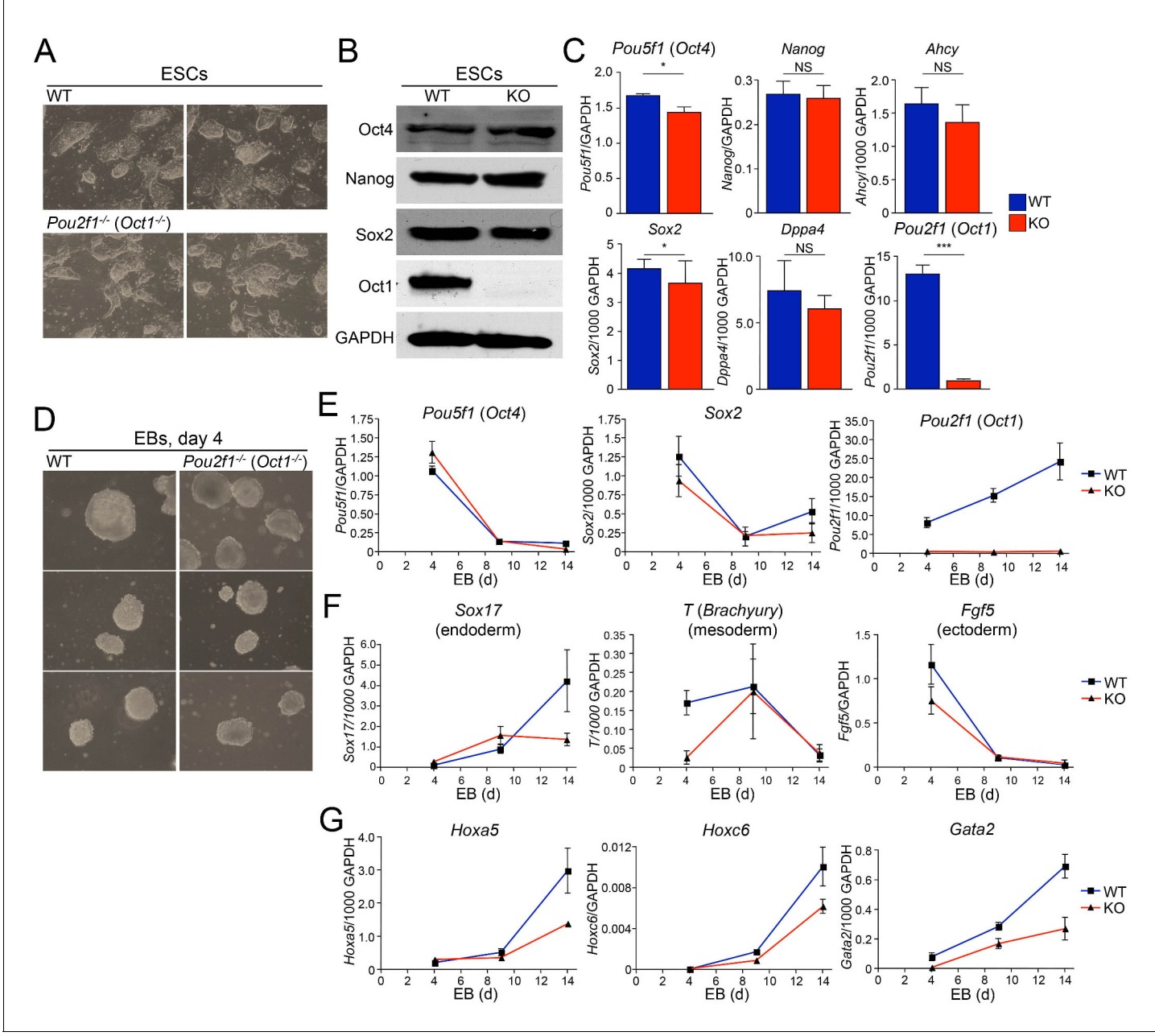

**Figure 1.** Abnormal developmental gene induction in ESCs lacking Oct1. (**A**) Phase microscopy images of four ESC lines (two Oct1 deficient, two WT littermate controls) derived from *Pou2f1*[-/+] intercrosses. Passage 5 ESCs on feeder fibroblasts are shown. (**B**) Immunoblot comparing lysates of a WT control line and littermate Oct1-deficient line. GAPDH is shown as a loading control. (**C**) mRNA expression of six genes in WT control and littermate Oct1-deficient ESC lines. Data were obtained by RT-qPCR using three biological replicates of a single line of each genotype. Error bars denote standard deviations. *p*-values: NS=non-significant, * < 0.05, ** < 0.01, *** < 0.001. (**D**) Phase microscopy images of 4-day EBs derived from ESCs ±Oct1. Three representative images of each genotype from wells of a 96-well plate are shown. (**E**) EBs were collected at 4, 9, and 14 days, and cDNA was prepared and subjected to RT-qPCR. Expression levels were normalized to GAPDH. Pluripotency genes (*Pou5f1, Sox2*) and *Pou2f1* were tested. Three biological replicates were performed. Error bars denote ±standard deviation. (**F**) Additional genes representative of all three germ layers, *Sox17*, *T*, and *Fgf5*, were tested as in E. (**G**) Three known poised Oct4 target genes, *Hoxa5*, *Hoxc6,* and *Gata2*, were tested as in E.

The following figure supplement is available for figure 1:

**Figure supplement 1.** Abnormal morphology in differentiating Oct1-deficient cells manifests by day 5 of EB formation.

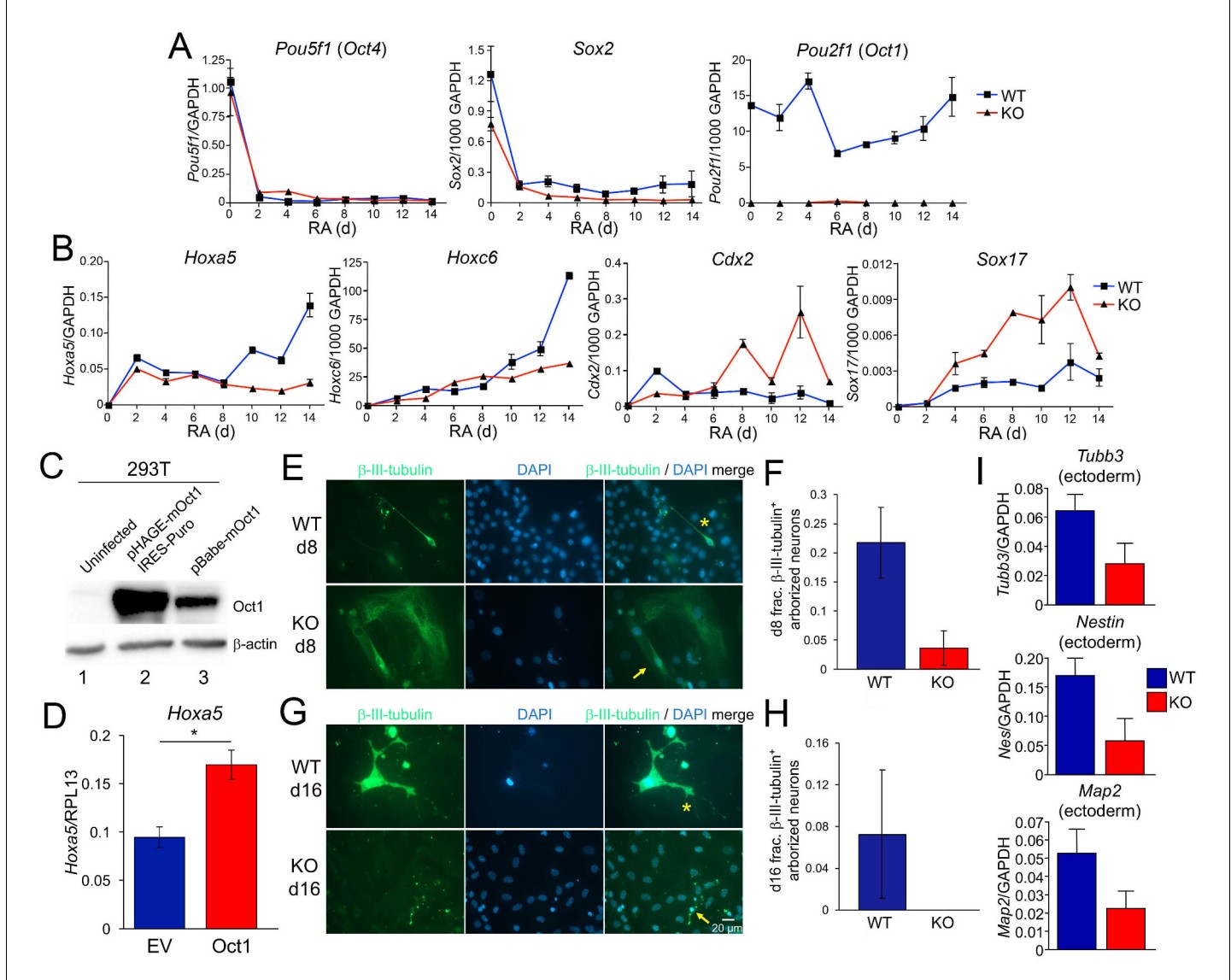

**Figure 2.** Effect of Oct1 loss on RA-mediated differentiation and neurogenesis. (**A**) Quantitative RT-PCR results are shown for *Pou5f1* (Oct4), *Sox2,* and *Pou2f1* (Oct1) mRNA relative to a GAPDH standard. Average of three biological replicates ±standard deviation is shown. (**B**) Similar analysis performed for the Oct4 targets *Hoxa5*, *Hoxc6*, *Cdx2*, and *Sox17*. (**C**) 293 T cells were transiently transfected with a lentiviral vector (pHAGE) expressing mouse Oct1. Lysates were prepared 48 hr later and immunoblotted for Oct1. Un-transfected cells are shown as a negative control (lane 1). Oct1 is not visible in these cells because of the lightness of the exposure. The same cells transduced with a retroviral vector encoding Oct1 (lane 3) are shown as a positive control. β-actin is shown as a loading control. (**D**) Oct1-deficient ESCs were differentiated using RA for 14 days. 4 days into the timecourse, cells were infected with lentiviruses expressing Oct1 and a puromycin resistance cassette, or an empty vector (EV) control. Cells were selected with puromycin for the remainder of the timecourse. cDNAs from the endpoint cultured were used to study expression of *Hoxa5* relative to an RPL13 ribosomal protein internal standard. Cells were prepared in triplicate for each condition. Error bars denote ±standard deviation. *denotes p<0.05. (**E**) Immunofluorescence images of WT and Oct1-deficient ESCs differentiated into neurons. Cells were cultured as EBs for 8 days, followed by culture for a further 8 days in neuralizing media (see Materials and methods). β-tubulin III and DAPI staining are shown. (**F**) Quantification of 300–400 cells from three individual differentiation experiments. Error bars denote ±standard deviation. (**G**) Similar to (**E**) except cells were cultured for eight additional d in neuralizing media. (**H**) Similar to (**F**) except using cells cultured for eight additional d. (**I**) 16 d-differentiated neuron cultures of similar genotypes were pooled and subjected to RT-qPCR using primers specific for *Tubb3*, *Nestin*, and *Map2*. Expression was assessed relative to GAPDH. Averages of three biological replicates are shown. Error bars denote ±standard deviation.

The following figure supplement is available for figure 2:

**Figure supplement 1.** *β*-tubulin III staining of neuralizing WT and Oct1-deficient EBs.

were infected over 2 consecutive days and selected with puromycin throughout the remained of the 14-day differentiation timecourse. At timepoints after day 6, infection and selection with empty vector skewed the expression of genes such as *Hoxa5*, suggesting that infection and selection were skewing the populations of cells in the culture. Cells infected at 4 and 5 days, however, did not show major differences (not shown), suggesting that the composition of cells in culture was not being significantly altered. We therefore infected differentiating Oct1-deficient cells consecutively on days 4 and 5, prepared cDNAs at day 14 and examined gene expression. By RT-qPCR, Oct1 was undetectable in cells transduced and selected with empty vector but robustly expressed by cells transduced with Oct1 (not shown). Expression of the developmentally-inducible *Hoxa5* gene was significantly augmented (p=0.026) by ectopic Oct1 expression (*Figure 2D*). These results indicate that restoration of Oct1 expression at these times and conditions can correct at least some of the gene expression defects associated with Oct1 deficiency.

RA-mediated differentiation yields neuronal precursor cells but not neurons. We used a differentiation system involving EB generation and culture in insulin, transferrin and selenium (see Materials and methods) to generate arborized neurons that express the marker β-tubulin III (*Tubb3*) and the neuroectoderm genes Nestin (*Nes*) and *Map2*. Staining of 2-day-old EBs for β-tubulin III prior to laminin/poly-L-lysine dish attachment – early in the differentiation protocol - revealed fewer β-tubulin III-positive cells in the Oct1 deficient condition (*Figure 2—figure supplement 1*). Upon complete differentiation (8 d EBs, 8 days in neuralizing monolayer culture), WT ESCs formed neurons robustly (*Figure 2E*, asterisk) while few β-tubulin III-expressing neurons were formed from Oct1-deficient ESCs. Oct1-deficient cells that did induce β-tubulin III tended to do so at lower levels, and the few cells that did express β-tubulin III robustly were nevertheless abnormal (*Figure 2E–F*, arrow). To test if Oct1 loss induced a kinetic delay that could be overcome by longer culture, cells were incubated for 8 or 16 additional days (16 or 24 days in neuralizing medium, 24 or 32 days total differentiation). In neither case were neurons formed (*Figure 2G–H* and data not shown). To study gene expression, individual wells of common genotypes differentiated for 16 days were pooled and subjected to RT-qPCR for *Tubb3*, *Nes* and *Map2*. Each of these genes showed defective expression in the absence of Oct1 (*Figure 2I*).

To test an unrelated developmental system, we performed cardiomyocyte differentiation by culturing EBs in hanging drops followed by culture with gelatin (see Materials and methods). Oct1-deficient ESCs failed to form beating cardiomyocytes, unlike WT (*Figure 3A* and *Videos 1–12*). RNA was collected from pooled beating and non-beating WT colonies, and Oct1-deficient colonies, and used to analyze *Mef2c* and *Hand1*, regulators of cardiomyocyte differentiation, and the terminal differentiation markers *Mlc2v* and *Mlc2a*. We observed *Mlc2v* expression defects in the Oct1-deficient condition equivalent to non-beating cardiomyocyte colonies from WT ESCs. *Mlc2a* and *Mef2c* expression was even weaker than non-beating WT cardiomyocyte colonies (*Figure 3B*). In contrast, *Hand1* showed no expression defects (*Figure 3B*), indicating that Oct1 deficiency does not globally down-regulate genes associated with cardiomyocyte differentiation. Cumulatively, the results indicate that although Oct1-deficient ESCs appear normal in the absence of differentiation cues, they do not induce poised developmentally inducible genes and fail to repress lineage-inappropriate genes such as *Cdx2* and *Sox17*, resulting in multiple cellular defects following differentiation.

## Oct1 conditional-deficient ESCs display abnormal gene expression upon differentiation

Although the ESC lines described above were derived from littermate animals and had normal karyotypes, it was possible that the developmental phenotypes and altered gene expression patterns resulted from differences unrelated to Oct1 status. Furthermore, the observed gene expression defects could result from compensatory changes due to development in an Oct1-deficient environment. Finally, the allele used to generate these lines is a severe hypomorph rather than a complete null (*Wang et al., 2004*). To circumvent these issues, we generated tamoxifen-inducible, Oct1 conditional-deficient ESCs.

We previously described *Pou2f1* conditional (floxed) mice (*Shakya et al., 2015b*). We generated inducible-conditional Oct1 ESCs by crossing the floxed allele onto *Rosa26*-Cre-ERT2 and *Rosa26*-lox-stop-lox-YFP (see Materials and methods). Pregnant animals were used to isolate *Pou2f1*^f1/fl^; *Rosa26*-Cre-ERT2;*Rosa26*-lox-stop-lox-YFP ESC lines in which Oct1 could be acutely deleted and YFP induced by 4-hydroxytamoxifen (4-OHT) administration. Treatment of parent ESCs with 4-OHT

**Figure 3.** Defective cardiomyocyte differentiation in ESCs lacking Oct1. (A) Cardiomyocytes were generated from individual EBs using 24-well dishes with gelatin. Functionality (±beating) was assessed for each well (16 per genotype) and plotted. (B) The wells assessed in (A) were pooled according to genotype and function (beating WT, non-beating WT and non-beating Oct1 deficient), cDNA was prepared and used for RT-qPCR using primers for *Mlc2v*, *Mlc2a*, *Mef2c* and *Hand1*. Averaged results from three replicates are shown. Error bars denote standard deviation. p-Values: NS=non-significant, * < 0.05, ** < 0.01, *** < 0.001.

resulted in variegated YFP⁺ colonies (*Figure 4A*, step one at top). Colonies with good morphology were picked (red arrow), trypsinized and expanded into derived ESC lines (*Figure 4A*, step two at

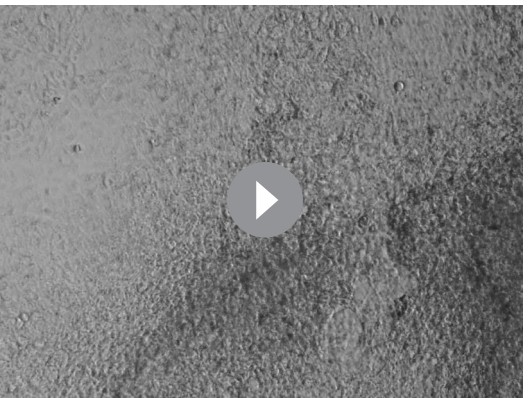

**Video 1.** Example WT ESC line cardiomyocyte differentiation 1.

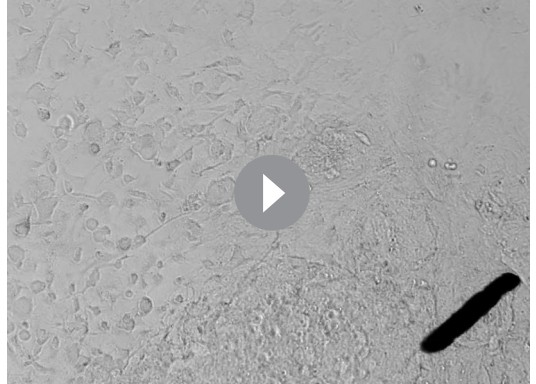

**Video 2.** Example WT ESC line cardiomyocyte differentiation 2.

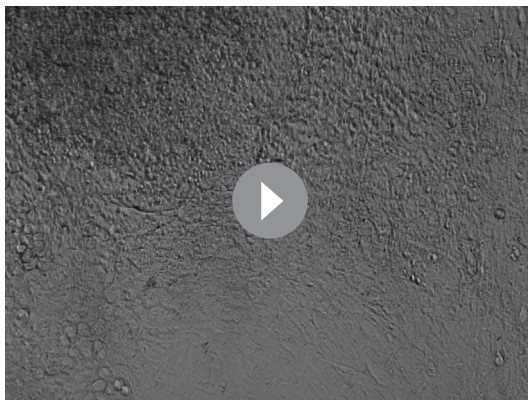

**Video 3.** Example WT ESC line cardiomyocyte differentiation 3.

bottom) that genotyped as $Pou2f1^{\Delta/\Delta}$ or $Pou2f1^{fl/\Delta}$ (**Figure 4B**). The designation $\Delta$ will be used to differentiate this allele from the germline deficient allele used in **Figures 1–3**. As with Oct1 germline-deficient ESCs, derived $Pou2f1^{\Delta/\Delta}$ ESC lines displayed normal colony morphology (**Figure 4C**), proliferated normally (not shown) yet expressed no Oct1 (**Figure 4D**). The derived cells showed normal karyotype profiles and could be propagated for >1 month without loss of an undifferentiated phenotype (not shown). Similar to germline Oct1 deficient ESCs, derived $Pou2f1^{\Delta/\Delta}$ cells also expressed Oct4, Sox2 and Nanog at normal levels (**Figure 4D**).

In differentiated cells, Oct1 promotes glycolysis and dampens mitochondrial function. Oct1 deficiency dramatically increases mitochondrial amino acid oxidation and oxygen consumption while decreasing glycolysis and to a lesser extent glucose oxidation (**Shakya et al., 2009**). These changes contribute to failure of fibroblasts to undergo oncogenic transformation, despite the fact that they grow at normal rates and can be immortalized by serial passage. To test if similar changes occur in ESCs lacking Oct1, we analyzed the metabolic profile of these cells. Few differences were noted, with only phosphoethanolamine (also known as phosphorylethanolamine, p=0.047) and inositol (p=0.037) showing significant changes (**Figure 4—figure supplement 1**). The lack of difference may be due to redundant functions of co-expressed Oct4 and Oct6, or co-selection for metabolic stability when selecting and propagating ESCs.

Derived $Pou2f1^{\Delta/\Delta}$ ESC lines, and parent cell line controls, were subjected to RA-mediated differentiation. Similar to results using Oct1 germline-deficient ESCs, the derived $Pou2f1^{\Delta/\Delta}$ ESCs lost Oct4 and Sox2 expression with kinetics identical to the parent line (**Figure 4E**). Microscopic imaging of the differentiating cells revealed that they were morphologically similar until approximately d 12, at which point $Pou2f1^{\Delta/\Delta}$ cells showed an increase in columnar/epithelial appearance (**Figure 4—figure supplement 2**). Also as before, the induction of silent, developmentally poised genes was defective: Hoxa5 and Hoxc6 both showed reduced expression in timecourse assays (**Figure 4F**). The cells also showed ectopic Cdx2 expression upon RA treatment (**Figure 4F**). As with germline-deficient ESCs, $Pou2f1^{\Delta/\Delta}$ ESCs did not generate true neurons efficiently (**Figure 4G**).

To determine the effect of conditional Oct1 loss during differentiation, parent cells were treated with 4-OHT following 8 d EB formation and 4 d in insulin, transferrin and selenium. After an additional 4 days, cells were fixed and stained with antibodies against β-tubulin III to score neurogenesis and YFP to score deletion. 40–50% of the treated cells induced YFP. Nearly all cells that induced β-tubulin III and/or generated neuron morphology lacked YFP expression (**Figure 4I** and **Figure 4J**). A few cells (2/ ~ 700) were both YFP- and β-tubulin III-positive (not shown), though it is possible that these cells are Pou2f1 heterozygous as 4-OHT treatment can result in recombination of only one allele (**Figure 4B**).

## Oct1-deficient ESCs form smaller, less differentiated teratomas and fail to generate chimeric mice

Parent and $Pou2f1^{\Delta/\Delta}$ ESCs were injected subcutaneously into contralateral flanks of NCr Nude immunocompromised animals to generate teratomas. ESCs lacking Oct1 consistently generated smaller tumors (**Figure 5A–C**). Immunoblotting confirmed that recovered tumors maintained their original Oct1 status (**Figure 5D**). Histological analysis confirmed that parent cells generated mature teratomas that included, e.g., glial tissue, and glandular epithelial and squamous elements (**Figure 5E**). In contrast, Oct1 deficient ESCs generated areas of focally immature cells, consistent with reduced differentiation. Occasionally tumors were comprised virtually entirely of primitive malignant cells resembling a germ cell tumor (**Figure 5E**, lower right).

A standard measure of pluripotency is the ability to contribute efficiently to adult cells and tissues (*De Los Angeles et al., 2015*). We injected parent and *Pou2f1*$^{\Delta/\Delta}$ ESCs into albino C57BL/6 blastocysts, resulting in high contribution in the case of the parent line (*Figure 5F*, left), but no contribution in the case of the derived lines (right). The average percent chimerism from two separate sets of injections (33 animals from parent cell line injections, 36 combined from two different *Pou2f1*$^{\Delta/\Delta}$ lines) confirmed the lack of contribution (*Figure 5G*). 18/33 animals injected with parent ESCs showed some detectable chimerism (55%), while 1/36 animals injected with conditional knockout ESCs showed transient trace chimerism in the eye (0.03%). The cells were imaged immediately prior to blastocyst injection to confirm an undifferentiated phenotype (*Figure 5—figure supplement 1*).

## Defects in lineage-specific gene expression in differentiated Oct1-deficient ESCs

To identify gene expression changes stemming from loss of Oct1, we performed RNAseq with undifferentiated and 14 d RA-differentiated parent and *Pou2f1*$^{\Delta/\Delta}$ ESCs. Three independent replicates were performed for each of the four conditions. Between 18.1 and 24.9 million sequence reads were generated for each sample, 73% to 82% of which aligned uniquely to the mouse *Mm10* reference genome. 99.6% of the reads within coding regions aligned to the correct strand. Variance between replicates was similar regardless of genotype or differentiation state (not shown). Unsupervised hierarchical clustering indicated that 0 and 14 days samples separated clearly from each other regardless of genotype, while within each timepoint the KO and parent WT samples clustered together (*Figure 6—figure supplement 1A*). These results indicated that the effect of RA treatment and differentiation on gene expression was far stronger than the effect of Oct1 deletion. Plotting gene expression levels in the parent vs *Pou2f1*$^{\Delta/\Delta}$ cells (*Figure 6A*) showed relatively few gene expression changes in the undifferentiated condition (>2.5 fold, p<0.01, 253 total genes). These genes never changed by >7 fold (*Supplementary file 1*). In contrast, 1123 genes change expression in differentiated *Pou2f1*$^{\Delta/\Delta}$ cells, some of which varied by >200 fold. Plotting gene expression fold change vs. *p*-value (*Figure 6B*) recapitulated these findings. Comparing genes differentially expressed at the two timepoints revealed little overlap (23 genes, *Figure 6C*). Analysis of the genomic alignments revealed that expression of control genes such as *Tbp* was unaltered, while pluripotency genes such as *Nanog* were silenced equivalently (*Figure 6D*). Other pluripotency genes such as *Pou5f1*, *Klf4*, *Dnmt3l*, and *Dppa4* behaved similarly to *Nanog* (not shown). One gene showing increased expression in the undifferentiated state was *Pou2f3* (Oct11, *Figure 6—figure supplement 1B*). *Pou2f3* shows low but detectable mRNA expression in WT ESCs, and is repressed upon differentiation to nearly undetectable levels. It is slightly elevated in Oct1-deficient ESCs but decreases to an even greater extent upon differentiation. RT-qPCR confirmed these changes in the context of overall low expression (*Figure 6—figure supplement 1C*). It is therefore unlikely that this protein provides a compensatory function upon differentiation.

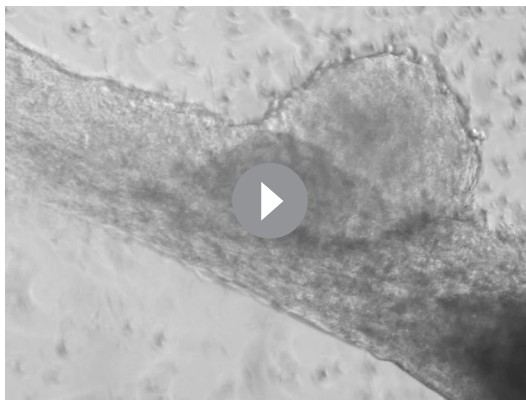

**Video 4.** Example WT ESC line cardiomyocyte differentiation 4.

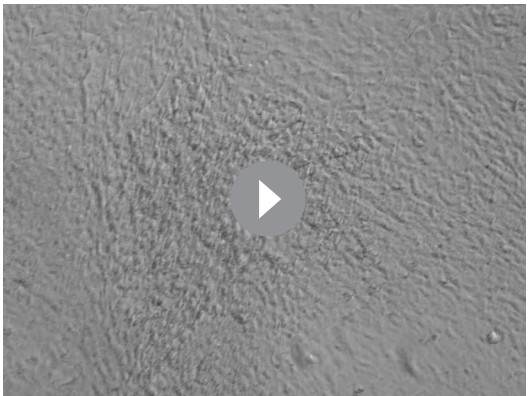

**Video 5.** Example WT ESC line cardiomyocyte differentiation 5.

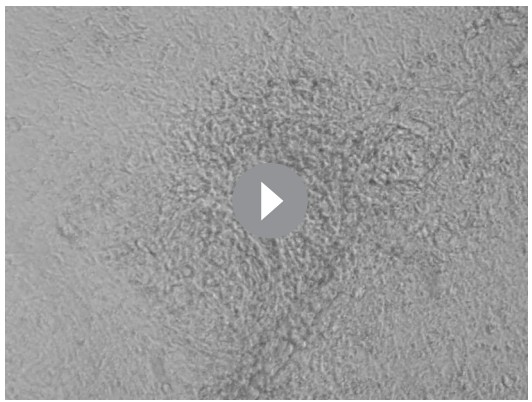
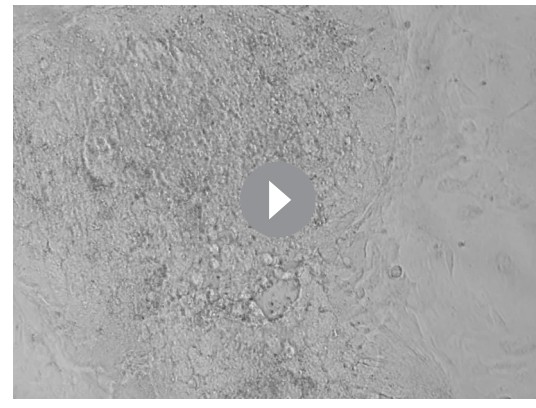

**Video 6.** Example WT ESC line cardiomyocyte differentiation 6.

**Video 7.** Example WT ESC line cardiomyocyte differentiation 1.

Approximately 300 genes were induced poorly in RA treated $Pou2f1^{\Delta/\Delta}$ cells relative to normal controls. Many of these encode regulators of neuronal specification and differentiation. Examples include *Hoxa5*, *Hoxb9* and *Nppa* (**Figure 6E**). Other examples include *Foxg1*, *Pcdh17*, *Ptgfr*, *Akr1c18*, *Cts7*, *Duox2*, *Hoxc5*, *Hoxc6*, *Hoxc8*, *Hoxc10,* and *Dcx* (**Figure 6—figure supplement 2A**). In addition, *Ahcy*, a stress-responsive Oct1 target (**Kang et al., 2009**; **Shakya et al., 2011**) showed weakened expression in the absence of Oct1 specifically in the differentiated condition (**Figure 6E**). An even larger cohort of ~800 genes was aberrantly expressed upon differentiation of $Pou2f1^{\Delta/\Delta}$ cells. These genes are strongly associated with alternative developmental fates (**Figure 6F** and **Figure 6—figure supplement 2B**). Examples include *Sox17*, *Cdx2* and *Gata4* (endoderm), *Fgb* (endoderm/liver), *Gata2* (mesoderm/endothelial), *Pparg* and *Irx3* (mesoderm/mesenchymal), *Muc13* (epithelial/hematopoietic) and *Tnfrsf9* (which encodes CD137/hematopoietic). The difference in *Sox17* and *Gata2* expression between EBs (defective) and RA-differentiated cells (elevated) likely arises from the additional developmental fates specified in EBs.

Unexpectedly, differentiating $Pou2f1^{\Delta/\Delta}$ cells also resulted in inappropriate expression of genes associated with trophoblast and placental development, the specification of which is normally restricted to trophectoderm cells rather than the inner cell mass (from which ESCs are derived). Examples include *Cdx2* (which is also expressed in endoderm), *Prl8a6*, *Hand1* (which is also expressed in cardiomyocytes), *Pappa2*, *Prl3b1*, and *Psg27* (**Figure 6F** and **Figure 6—figure supplement 2B**). Some of these genes are also expressed in other lineages while others are highly specific. In aggregate, they indicate improper activation of an extra-embryonic program. Using RT-qPCR we confirmed unperturbed expression of *Tbp*, defective expression of *Ahcy*, and elevated expression of *Prl8a6* in differentiating Oct1-deficient ESCs (**Figure 6G**). These results indicate that Oct1 deficiency results in defective lineage specification upon differentiation.

We used ChIPseq to identify common and unique Oct1 and Oct4 target genes in ESCs. We also performed H3K4me3 ChIPseq as a control. The ChIPseq data were of high quality based on measures of signal/noise ratio (see Materials and methods). After filtering, 27.3 (Oct1), and 23.7 (Oct4) million alignable reads were generated, corresponding to 692 (Oct1), and 8673 (Oct4) peaks. Allocating the peaks to nearest genes revealed 209 unique Oct1 target genes, 356 common targets, and 5563 unique Oct4 targets (**Figure 7A**). The smaller size of the Oct1 target

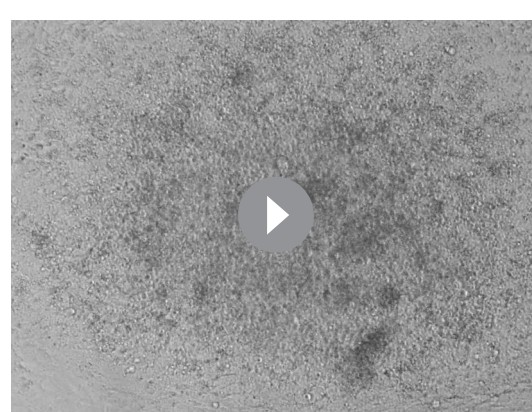

**Video 8.** Example WT ESC line cardiomyocyte differentiation 2.

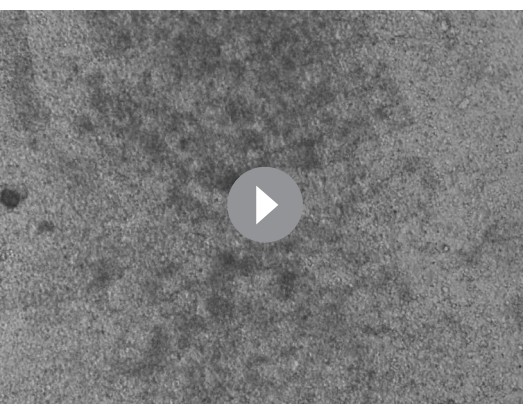

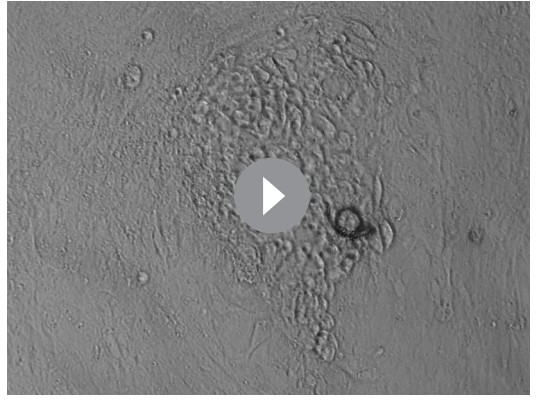

**Video 9.** Example WT ESC line cardiomyocyte differentiation 3.

**Video 10.** Example WT ESC line cardiomyocyte differentiation 4.

pool relative to Oct4 may be attributable to >10 fold lower Oct1 levels in ESCs as observed by RT-qPCR (*Figure 1C*, *Figure 1E*, *Figure 2A*) and RNAseq (not shown). Oct1 may also require a more open chromatin context, and/or the presence of specific co-bound factors, to access DNA. For example *Il2* and *Ifng* are known Oct1 targets in differentiated T cells (*Shakya et al., 2015b*, *2011*) but were not identified as targets in this analysis.

Motif analysis of unique and co-bound peaks revealed significant differences in recognized DNA elements. Regions associated exclusively with Oct4 were significantly enriched for Oct-Sox compound elements that likely also associate with Sox2 in ESCs (*Figure 7B*). In contrast, target regions preferentially associated with Oct1 were enriched for the simple octamer element ATTTGCAT (shown by the software as an Oct4 motif in *Figure 7B*). Interestingly, co-occupied peaks strongly associate with a motif termed a MORE that is known to bind two Oct protein molecules (*Reményi et al., 2001*; *Tomilin et al., 2000*). In differentiated cells lacking Oct4, oxidative stress induces homodimeric Oct1 binding to MORE-containing genes such as *Polr2a*, *Ahcy*, *Ell,* and *Rras2*. Oxidative stress-induced binding occurs via phosphorylation of a conserved serine residue in the DNA-binding domain (*Kang et al., 2009*). These genes were constitutively co-bound by Oct1 and Oct4 in ESCs (*Figure 7—figure supplement 1*). Additional examples of genes associated with Oct4 alone (*Pou5f1*), or Oct1 alone (*Taf12*) are shown in *Figure 7C*. This panel also shows another example of a MORE containing gene (*Polr2a*, two tandem MOREs binding four molecules) that also associates with both proteins but shows an Oct1 bias, as well as an example (*Pax6*) that is bound by both proteins but in two different locations. Using ChIP-qPCR we validated two genes, *Polr2a* (Oct1-enriched) and *Pou5f1* (Oct4-enriched, *Figure 7D*). The complete set of identified targets is shown in *Supplementary file 2*.

Intersecting the ChIPseq and RNAseq data revealed little overlap. Only 34 Oct1-bound or Oct1/Oct4 co-bound targets showed differential expression following RA-mediated differentiation (*Figure 7A*). Examples include *Pank4*, *Cdh5* and *Med16*. 193 Oct1-bound and 325 Oct1/Oct4-co-bound genes did not show expression differences at d 14. Examples include *Tbx3*, *Tcf4*, and *Txb6*, which also showed no differences throughout the differentiation timecourse (*Figure 7E*). Instead, 1066 genes with altered expression in differentiated Oct1-deficient cells showed Oct4 but not Oct1 enrichment. These findings indicate that (1) identified Oct1 targets were not differentially expressed upon

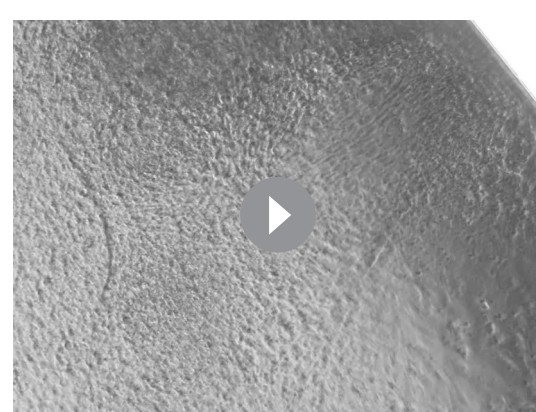

**Video 11.** Example WT ESC line cardiomyocyte differentiation 5.

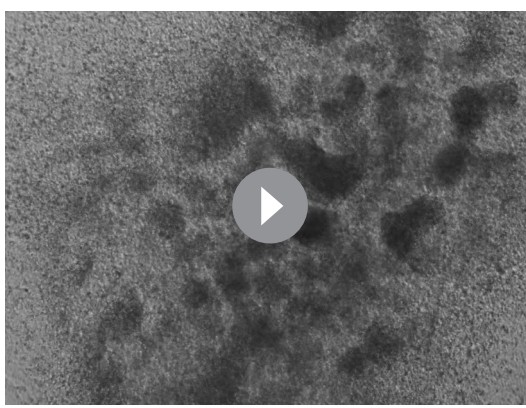

**Video 12.** Example WT ESC line cardiomyocyte differentiation 6.

differentiation, and (2) developmental genes shown to be differentially expressed in RA treated Oct1-deficient cells were not Oct1 targets in ESCs.

The above findings could be reconciled by postulating that (1) Oct functions at co-bound targets in ESCs to buffer them against oxidative stress as described previously in fibroblasts (*Kang et al., 2009*; *Shakya et al., 2011*), and (2) developmental genes that are differentially expressed in Oct1-deficient cells but exclusively bound by Oct4 in ESCs become Oct1 targets during the differentiation process as Oct4 is lost. To test the first hypothesis, we studied the effect of $H_2O_2$ exposure on the expression of two co-bound genes, *Ahcy* and *Polr2a*, in ESCs ±Oct1. Both genes contain conserved MORE sequences (*Figure 8A*). Treatment of cells with 2 mM $H_2O_2$ resulted in a rapid loss of *Ahcy* and *Polr2a* mRNA specifically in Oct1-deficient ESCs (*Figure 8B*), exactly as observed in fibroblasts (*Kang et al., 2009*; *Shakya et al., 2011*). As expected, these cells were hypersensitive to $H_2O_2$ (*Figure 8C*). These results suggest that as in other cell types, Oct1 functions in ESCs to buffer these genes from oxidative stress-associated inhibition.

To test the hypothesis that Oct1 occupies Oct4 targets as cells differentiate and Oct4 is lost, we performed ChIP-qPCR timecourses using differentiating ESCs and antibodies against Oct1 and Oct4. Material was collected from 0, 2, 4, 6, 8, 10, 12 and 14 d of differentiation with RA. We chose a gene, *Hoxc5*, that contains a conserved perfect octamer sequence (*Figure 8D*), but is not an Oct1 target based on ChIPseq (*Figure 8E*). *Hoxc5* also shows poor induction in upon RA-mediated differentiation of $Pou2f1^{\Delta/\Delta}$ cells (*Figure 6—figure supplement 2A*). Oct1 ChIP-qPCR revealed no binding in ESCs, as expected based on the ChIPseq (*Figure 8F*), however robust binding was transiently observed at 6 d. By 14 d of differentiation Oct1 binding was again undetectable. We also examined a target region between the linked *Myf5* and *Myf6* (*Mrf4*) loci on chromosome 10 that contains several near-perfect octamer sites (not shown), and is strongly bound by Oct4 but not Oct1 (*Figure 8E*). Oct1 inducibly occupied this region even more rapidly (2 d) as Oct4 binding was lost, and in this case Oct1 binding was maintained, at varying levels, during ESC differentiation (*Figure 8F*). Finally, we studied two additional genes, *Pou5f1* and *Rest* (Nrsf), which also contain conserved perfect octamer sequences in their regulatory regions and also show exclusive Oct4 binding. These genes are both expressed in ESCs and silenced as differentiation proceeds. Here early Oct1 binding was identified, which was maintained at low levels during the differentiation timecourse in the case of *Pou5f1* but transient in the case of *Rest* (*Figure 8F*). These results indicate a highly dynamic interplay between Oct1 and Oct4 in differentiating cells.

## Discussion

Our results indicate that Oct1-deficient ESCs are unperturbed in terms of morphology, growth, metabolism, and gene expression. EBs formed from these cells are microscopically normal at early timepoints and express genes associated with all three germ layers. However, Oct1-deficient ESCs show phenotypic and molecular defects upon differentiation. These cells fail to form neurons and cardiomyocytes, generate smaller and less differentiated teratomas, and fail to contribute to adult mouse tissues. Prior work has shown that partial knockdown of Oct1 also inhibits neuron formation in the context of knockout of the related protein Oct2 (*Theodorou et al., 2009*).

Three molecular defects manifest upon differentiation of Oct1-deficient ESCs. First, loss of Oct1 results in a failure to fully induce genes associated with a given developmental lineage. Second, Oct1 is necessary for the repression of alternative embryonic developmental lineages. As a result, upon differentiation gene expression programs are marked not only by poor induction of lineage-appropriate gene expression, but also by ectopic expression of genes specific to alternative lineages

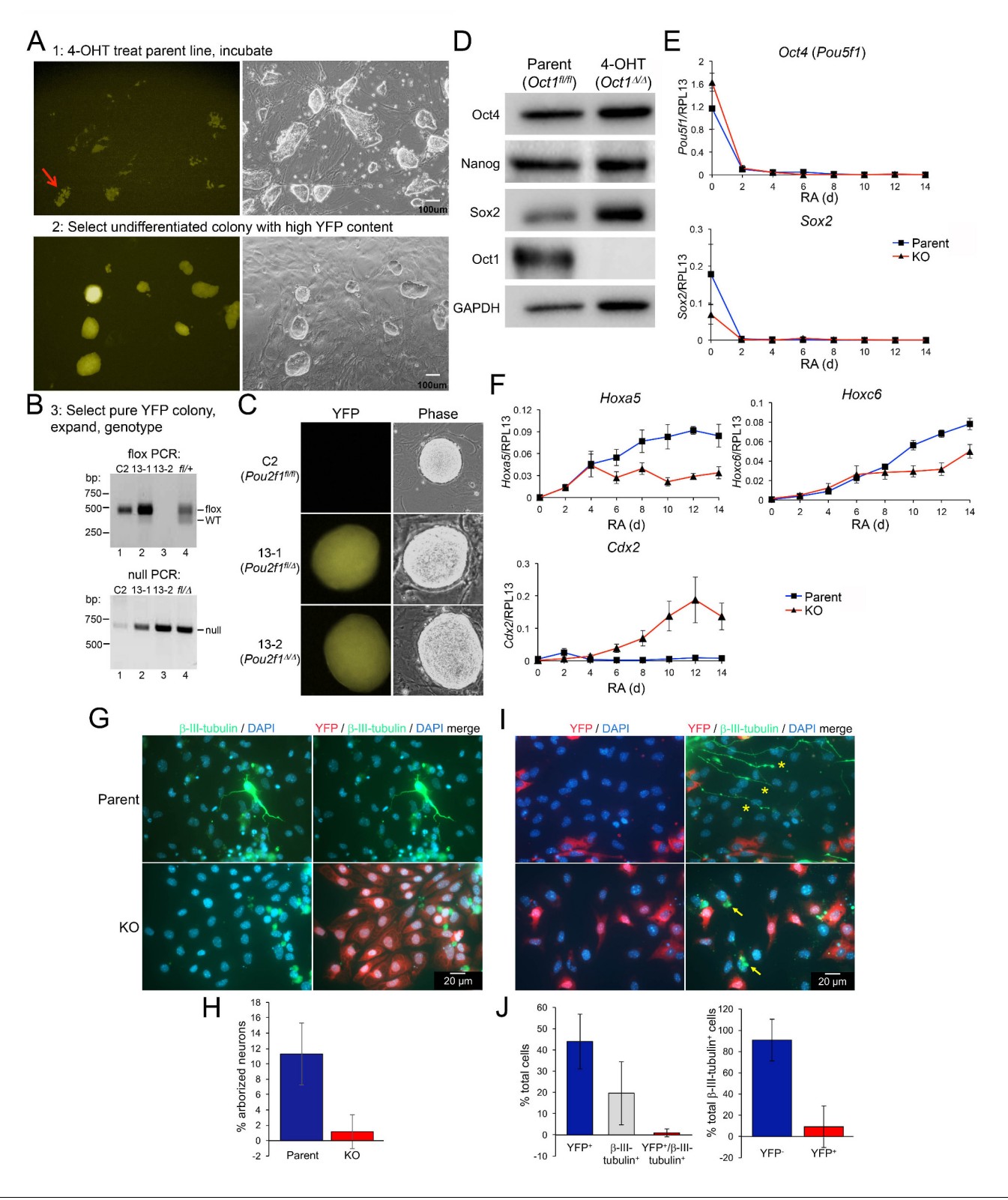

**Figure 4.** Gene expression defects upon differentiation of Oct1 inducible-conditional ESCs. (**A**) YFP-epifluorescence and phase microscopy images of inducible-conditional ESCs. Top: parent *Pou2f1*[fl/fl] cells were treated with 500 nM 4-OHT for 24 hr. A colony with good morphology and variegated YFP expression was picked, trypsinzed, replated and expanded. Bottom: derived *Pou2f1*[Δ/Δ] ESCs. (**B**) PCR genotyping of parent (C2) and derived (13-1, 13-2) lines. Feeder fibroblasts were depleted by two serial 1 hr platings on plastic. The residual WT band in lane two is due to feeder contamination. The
*Figure 4 continued on next page*

*Figure 4 continued*

rightmost lane (lane 4) shows control tail DNA from a *Pou2f1^{fl/+}* (top) or *Pou2f1^{fl/Δ}* animal. (C) Epifluorescence and phase microscopy images of single colonies. Images were taken at the same magnification. (D) Immunoblots comparing lysates of vehicle-treated parent ESCs and derived KO cells. GAPDH is shown as a loading control. (E) Feeder-depleted ESCs were treated continuously with RA on gelatin-coated plates in the absence of LIF for 14 days. Media was changed every other day. cDNA was prepared every other day and used in RT-qPCR with primers against *Pou5f1* and *Sox2*. Averages of three biological replicates ±standard deviation are shown. Methodologically, the experiments were performed identically to *Figure 2A–B*. (F) Additional RT-qPCR using primers against the Oct4 target genes *Hoxa5*, *Hoxc6* and *Cdx2*. (G) *Pou2f1^{Δ/Δ}* ESCs were differentiated into neurons as in *Figure 2C–G* by forming EBs for 8 days followed by 8 day in medium containing insulin, transferrin and selenium. Cells were fixed and used for immunofluorescence using DAPI and antibodies against YFP and β-tubulin III. (H) Immunofluorescence images were quantified based on arborized morphology. Approximately 700 total cells were analyzed. (I) Similar to (G) except parent *Pou2f1^{fl/fl}*, ESCs were used and after 4 days in culture cells were treated with 500 nM 4-OHT for 24 hr to delete Oct1 and induce YFP. Two representative images are shown. (J) Immunofluorescence images were quantified based on YFP and β-tubulin III positivity. Percent total cells showing single or double staining, or percent β-tubulin III^+ cells with and without YFP are shown. Approximately 700 total cells were analyzed.

The following figure supplements are available for figure 4:

**Figure supplement 1.** Steady-state metabolite levels in parental WT and 4-OHT-treated *Pou2f1^{Δ/Δ}* ESCs as determined by GC-MS.

**Figure supplement 2.** Images of parental WT and 4-OHT-treated KO ESCs differentiating in the presence of RA.

(*Figure 8G*). The third defect is mis-expression of genes associated with extra-embryonic lineages. These genes are normally under tight repression in ESCs and their differentiated progeny. For example, *Prl8a6* and *Prl3b1* are mis-expressed in RA-differentiated ESCs. Other genes such as *Cdx2* and *Hand1* are examples of genes expressed both in extra-embryonic and alternative embryonic lineages. For example, the *Cdx2* promoter contains a perfect consensus octamer element and is a known Oct1 target in somatic cells (*Jin and Li, 2001*; *Wang et al., 2009*). In the early embryo, *Cdx2* promotes trophectoderm fate and is under tight repression by Oct4 (*Yeap et al., 2009*; *Yuan et al., 2009*). Later in development, *Cdx2* is induced in the endoderm-derived developing gastrointestinal tract (*Guo et al., 2004*; *Lu et al., 2008*) and during primitive hematopoiesis (*Wang et al., 2008*), but is not widely expressed in ectoderm (*Suh and Traber, 1996*). In RA-differentiated cells, *Cdx2* thus represents both a lineage-inappropriate gene and an extra-embryonic lineage. *Cdx2* is mis-expressed following RA-mediated differentiation of both germline and inducible-conditional Oct1-deficient ESCs. Interestingly Oct1 may execute the opposite function in extra-embryonic tissue, as germline Oct1-deficient mice show defects in extra-embryonic tissues including poor expression of *Cdx2* (*Sebastiano et al., 2010*).

ChIPseq experiments reveal that Oct1 and Oct4 regulate common and distinct targets in ESCs. These differences in bound targets lead to functional consequences, as the two proteins recruit different cofactors such as Jmjd1a in the case of Oct1 and Jmjd1c in the case of Oct4 (*Shakya et al., 2015a*, *2011*). Oct4 occupies a large group of >5000 genes, including developmentally poised genes such as *Hoxc5* and *Myf5*, and core pluripotency genes such as *Pou5f1* and *Nanog*. Oct1 does not occupy these genes in ESCs, consistent with the ability of Oct1-deficient ESCs to maintain pluripotency. Instead Oct1 co-occupies a cohort of 325 genes with Oct4 that are highly enriched for a motif known as a MORE (*Reményi et al., 2001*; *Tomilin et al., 2000*). Oct proteins are known to homo- and hetero-dimerize (*Kang et al., 2009*; *Tantin et al., 2008*; *Tomilin et al., 2000*; *Verrijzer et al., 1992*). The configuration of Oct proteins can determine cofactor association and hence regulatory output (*Reményi et al., 2001*; *Tomilin et al., 2000*). Many of these constitutively co-bound genes were previously shown to become occupied by Oct1 upon oxidative stress exposure in differentiated cells lacking Oct4 (*Kang et al., 2009*). The function of Oct1 at these genes is to insulate them against inhibition by oxidative stress. Fibroblasts lacking Oct1 show inappropriate repression of MORE-containing genes following H₂O₂ exposure (*Kang et al., 2009*; *Shakya et al., 2011*). We demonstrate the identical phenotype using Oct1-deficient ESCs. Oct1 also exclusively associates with a small number (~200) of other genes including *Taf12*, which contains another binding site variant known as a TMFORE (*Kang et al., 2009*).

Notably, in undifferentiated cells Oct1 does not associate with developmental targets that become deregulated upon differentiation of Oct1-deficient ESCs. Oct4 is present at higher levels in

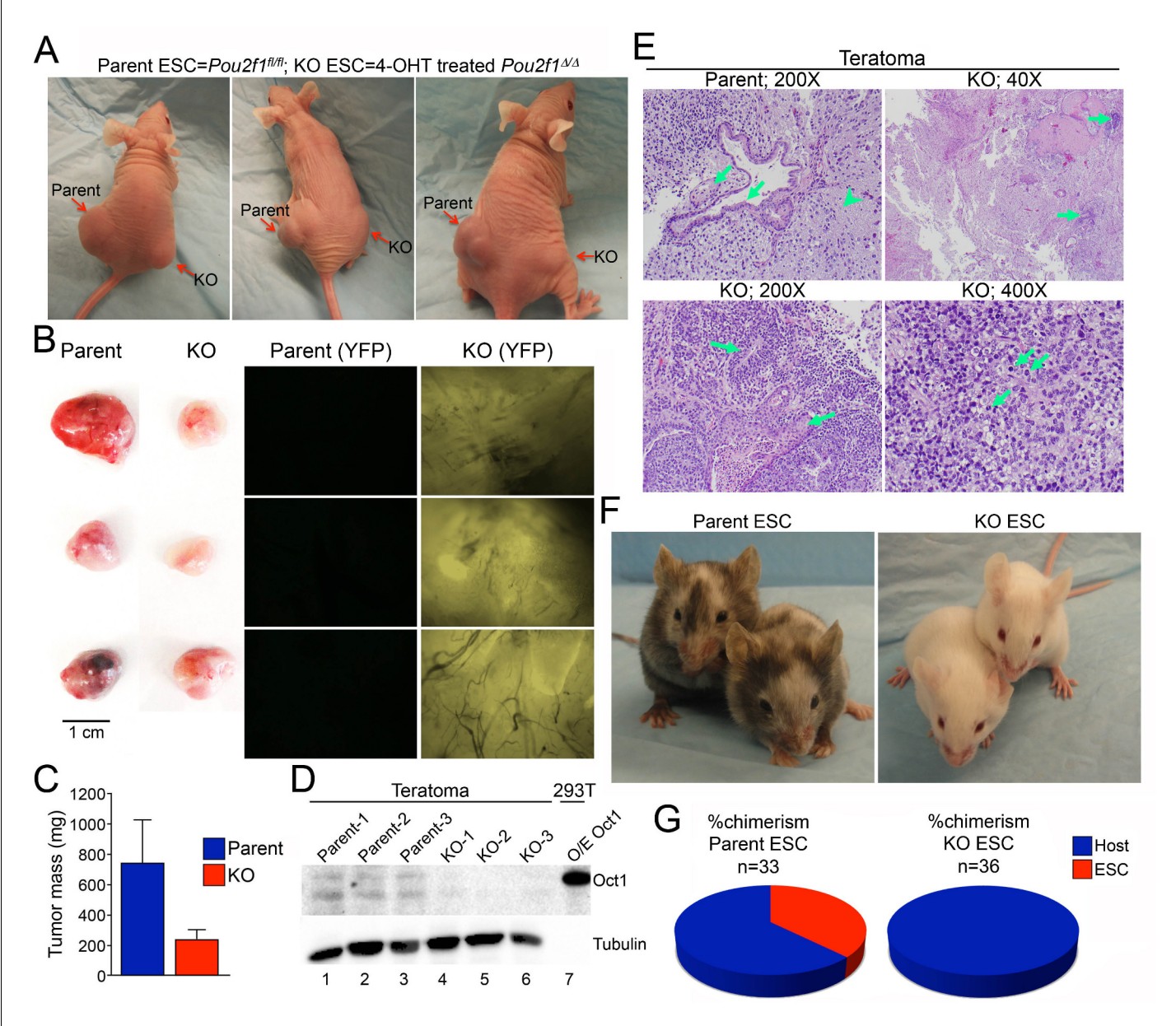

**Figure 5.** Smaller, less differentiated teratomas and lack of contribution to adult mouse tissues in *Pou2f1*$^{Δ/Δ}$ ESCs. (**A**) 1 × 10$^6$ ESCs were injected into flanks (left flank: control *Pou2f1*$^{fl/fl}$ parent cells, right flank: derived *Pou2f1*$^{Δ/Δ}$ ESCs) of NCr Nude mice. Images are shown at 4 weeks. (**B**) Images of dissected teratomas. Left side: white light. Scale in cm shown on the right. Right side: YFP fluorescence. (**C**) The mass from ten tumors was averaged and plotted. Error bars denote standard deviation. (**D**) Immunoblots are shown for Oct1 in lysates prepared from part of the individual teratomas. Lane 7: extracts from 293 T cells transiently over-expressing recombinant Oct1. α-Tubulin is shown as a loading control. The lack of α-Tubulin in lane seven arises from the fact that less protein was loaded due to high levels of recombinant Oct1. (**E**) H and E images of parent and KO teratomas. Top left: normal teratoma morphology comprised of mature elements, e.g glial tissue (arrowhead), mature squamous (left arrow) and ciliated glandular epithelial (right arrow). Top right: teratoma comprised predominantly of mature elements, but with immature elements (approximately 5% of the tumor, arrows). Bottom left: teratoma with both mature and immature elements represented. Mature squamous epithelium (right arrow) is abundant, while immature neuroepithelium (left arrow) is noted focally. Bottom right: tumor comprised almost entirely of a primitive malignant neoplasm that does not recapitulate any recognizable line of differentiation. Arranged in sheets and irregular nests, these cells exhibit marked cytologic atypia, with nuclear pleomorphism and coarsely-clumped chromatin. Nucleoli are variably prominent. Mitotic activity is brisk (arrows). (**F**) Parent ESCs (left side) or derived KO cells (right side) were injected into albino C57BL/6 blastocysts and implanted in pseudo-pregnant animals. Representative images are shown. (**G**) Average contribution is shown for the two cell types. ESCs contribution was assessed subjectively based on dark coat and eye color. 33 animals were tested in the case of the parent line and 36 animals were assessed in the case of the derived *Pou2f1*$^{Δ/Δ}$ line.

*Figure 5 continued on next page*

*Figure 5 continued*

The following figure supplement is available for figure 5:

**Figure supplement 1.** Pluripotent phenotype of ESCs immediately prior to blastocyst injection.

ESCs compared to Oct1, suggesting that mass action may contribute to the lack of Oct1 binding. This model predicts that Oct1 would occupy these genes as Oct4 is lost during differentiation. We tested four regions bound by Oct4 but not Oct1 in ESCs, *Hoxc5*, *Myf5/Myf6*, *Rest* and *Pou5f1*, predicting that Oct1 binding will manifest as cells differentiate and Oct4 is lost. In all cases, Oct1 binding was observed at one or more points during the differentiation timecourse. We propose that Oct1 transiently replaces Oct4 at many such Oct4 target genes upon differentiation, where it promotes lineage-appropriate target gene expression, and represses expression of lineage-inappropriate targets. The binding events occur during a brief but important window during which critical decisions about suppression or potentiation of lineage-specific developmental Oct4 target genes are made. Binding also occurs before many of the affected target genes are induced, suggesting that Oct1 is not the principal driver of expression of these genes, but instead establishes a chromatin context in which these genes remain poised for expression, or become permanently repressed. Of the genes tested in ChIP-qPCR RA differentiation timecourses, the lineage-appropriate *Hoxc5* gene shows poor induction in upon differentiation of Oct1-deficient cells (**Figure 6—figure supplement 2A**), *Myf5* and *Myf6* are mesoderm-specific and lineage-inappropriate, *Rest* is both pluripotency-associated and lineage-inappropriate, and *Pou5f1* is more restricted to ESCs. These latter genes showed no evidence of ectopic expression. This observation can be reconciled with our model by positing that redundant mechanisms, perhaps mediated by other Oct proteins such as Oct6/Pou3f1, enforce their repression in differentiating ESCs.

The bipotential function of Oct1 is consistent with previous findings in fibroblasts and T cells (**Shakya et al., 2011**). Oct1 functions are mediated in part through association with the inhibitory chromatin remodeling complex NuRD (**Shakya et al., 2011**), or with Jmjd1a/KDM3A, a histone H3K9me2 lysine demethylase (**Shakya et al., 2015b, 2011**). H3K9me2 also controls developmental gene induction (**Wen et al., 2009**; **Zylicz et al., 2015**) and reprogramming to pluripotency (**Chen et al., 2013**; **Sridharan et al., 2013**).

The ability of Oct1 to suppress genes for alternative developmental lineages is reminiscent of findings using T cells in which Oct1 suppresses alternative T cell lineage genes via inter-chromosomal communication between gene loci that execute opposing gene expression programs (**Kim et al., 2014**). Oct1 interacts with CTCF (**Kim et al., 2014**), helping it foster exclusive gene expression programs in T cells. More work is required to determine if Oct1 insures mutually exclusive embryonic developmental gene expression programs through similar mechanisms.

## Materials and methods

### Derivation of Oct1 germline and conditional ESCs

All mice were C57BL/6J background. Oct1 germline-deficient ESCs were generated by intercrossing heterozygous *Pou2f1*$^{-/+}$ mice (**Wang et al., 2004**) to generate a 1:2:1 ratio of *Pou2f1*$^{-/-}$: *Pou2f1*$^{-/+}$: *Pou2f1*$^{+/+}$ embryonic offspring. ESCs were derived from preimplantation blastocysts and genotyped. Heterozygous ESCs were not studied further. Littermate WT ESCs lines constituted the controls for these experiments. Oct1 inducible-conditional ESCs were generated by first separately crossing mice with the *Pou2f1* conditional (floxed) allele (**Shakya et al., 2015b**) to the YFP reporter B6.129 × 1-*Gt(ROSA)26Sor*$^{tm1(EYFP)Cos}$/J (Jackson labs #006148) and inducible cre transgenic line B6.129-*Gt(ROSA)26Sor*$^{tm1(cre/ERT2)Tyj}$/J (Jackson labs #008463). Resulting *Pou2f1*$^{fl/fl}$ animals were intercrossed to generate embryonic *Pou2f1*$^{fl/fl}$ offspring in which LSL-YFP was expressed from one *Rosa26* allele and Cre-ERT2 was expressed from the other. Parent ESCs were derived from these preimplantation blastocysts. The parent lines constituted the controls for derived 4-OHT-treated, *Pou2f1*$^{Δ/Δ}$:YFP$^+$ lines. Cell lines were routinely authenticated by genotyping. Mycoplasma testing was conducted regularly in-house using a previously published method (**Molla Kazemiha et al., 2009**). Cells were negative throughout the study.

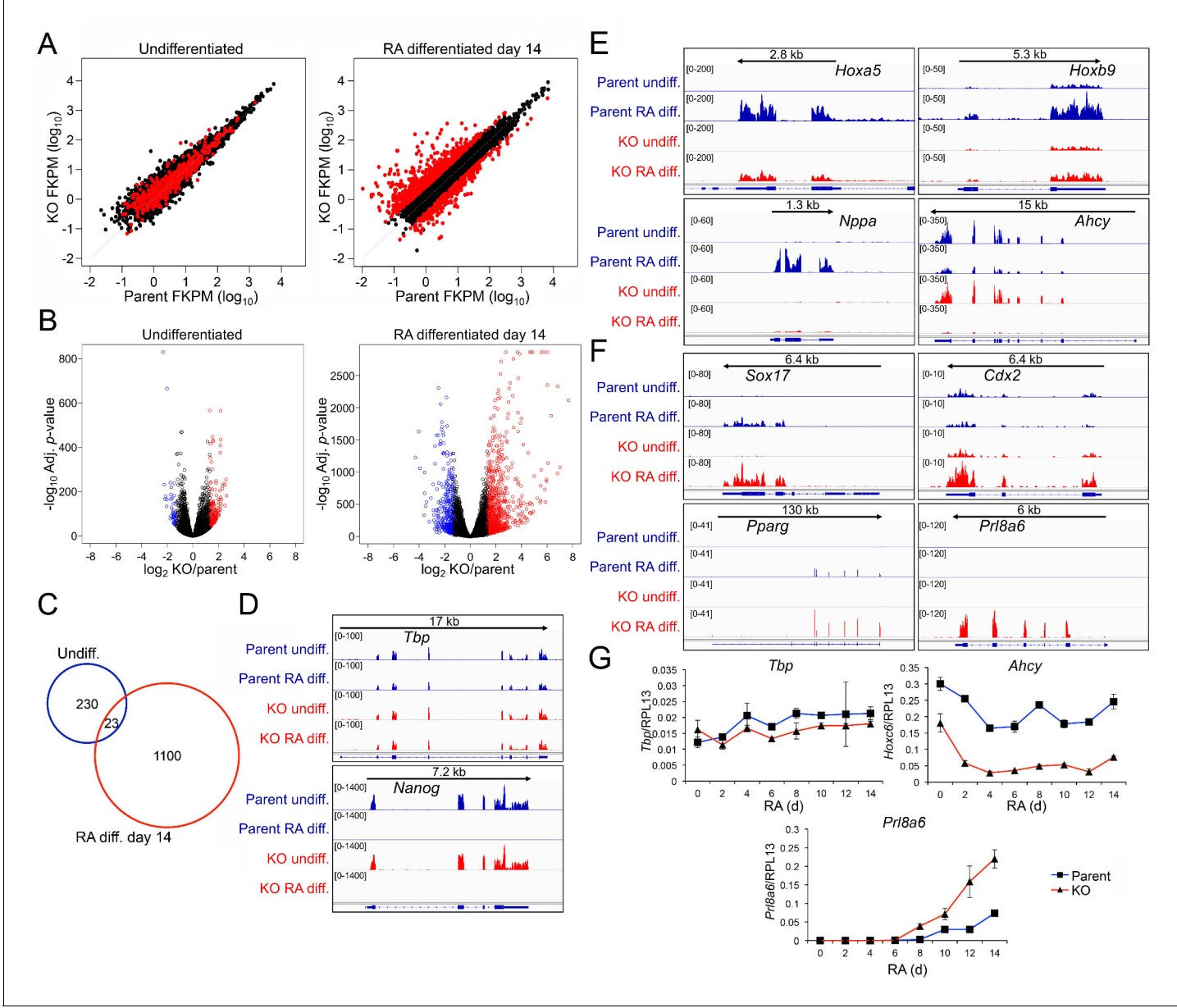

**Figure 6.** Genome-wide changes in developmental gene expression following differentiation of Oct1 conditional-inducible deficient ESCs. (**A**) For each gene, averaged RNAseq FKPM (aligned Fragments Per Kilobase per Million aligned reads) values from three undifferentiated or 14-day RA-differentiated parent and KO ESCs were plotted on a log₁₀ scale. Red genes signify significantly changed gene expression (Adj. p<0.01, fold change >2.5) at d 14. For each timepoint, genes showing <50 total reads in both genotypes conditions were called unexpressed and are not displayed. (**B**) Volcano plots showing log₂ averaged difference in gene expression vs. −10×log₁₀ significance. Significantly altered genes (Adj. p<0.01, 2.5-fold change) are shown in blue (down-regulated) or red (up-regulated). (**C**) Venn diagram showing total numbers of significantly (p<0.01) differentially expressed (>2.5 fold) genes in undifferentiated and 14-day RA-differentiated Oct1-deficient ESCs. Overlap shows genes differentially expressed at both timepoints. (**D**) Genome tracks of averaged RNAseq read densities (genome build mm10) for two control genes: *Tbp* (a constitutively expressed gene), and *Nanog* (expressed in pluripotent but not differentiated conditions). Arrows show directionality of gene transcription and size of the transcription unit. (**E**) Additional genome tracks are shown of three genes with poor induction in the KO condition: *Hoxa5*, *Hoxb9*, and *Nppa*. *Ahcy* is also shown, which becomes more strongly down-regulated in the differentiated condition. *Hoxa5* physically overlaps with *Hoxa3*, *Hoxaas3*, and *Hoxa6*, which are not highlighted. (**F**) Additional genome tracks are shown of genes showing ectopic expression in the differentiated condition: *Sox17*, *Cdx2*, *Pparg*, and *Plr8a6*. (**G**) RT-qPCR validations of additional genes identified by RNAseq, *Ahcy* and *Prl8a6*. *Tbp* is shown as a control. Average of three biological replicates ±standard deviation is shown.

The following figure supplements are available for figure 6:

*Figure 6 continued on next page*

*Figure 6 continued*

**Figure supplement 1.** Differences in gene expression in differentiated Oct1 deficient cells revealed by RNAseq.

**Figure supplement 2.** Differences in gene expression in differentiated Oct1 deficient cells revealed by RNAseq.

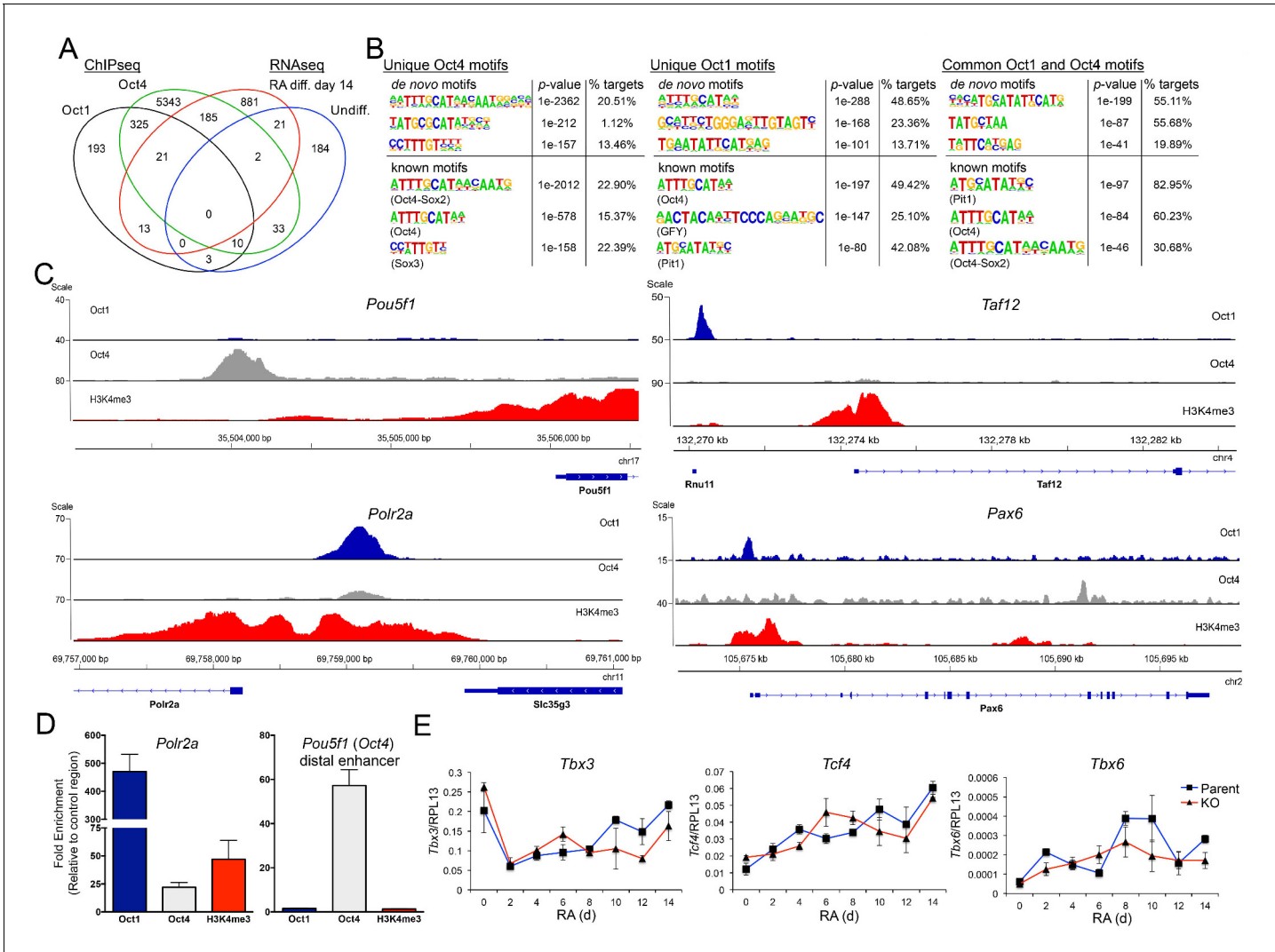

**Figure 7.** Unique and common Oct1 and Oct4 targets in ESCs. (**A**) Venn diagram illustrating Oct1 and Oct4 target gene profile and intersection with RNAseq gene set. (**B**) Motif analysis for peaks occupied uniquely by Oct1 or Oct4, and for peaks occupied by both proteins. Top shows best consensus sequences associated with binding. Bottom shows best matches to annotated weight matrices. In the case of known motifs, deviation of physiological binding sites from consensus causes recurring sequences meet threshold criteria for the compound 'Oct4-Sox2' site but not for a simple octamer site ('Oct4'). This is why the percentage of target sites computationally associated with 'Oct4-Sox2' is higher than for 'Oct4.' (**C**) Genome tracks showing ChIPseq enrichment for Oct1 (blue), Oct4 (gray) or H3K4me3 (red). Target gene and orientation is shown at the bottom of each track. *Pou5f1*, *Polr2a*, *Taf12* and *Pax6* are shown. (**D**) ChIP-qPCR validation of select ChIPseq targets. Fold enrichments using Oct1 and Oct4 antibodies at *Polr2a* and *Pou5f1* are shown. (**E**) RT-qPCR for three identified Oct1 target genes, *Tbx3*, *Tcf4* and *Tbx6* at 0, 2, 4,6, 8, 10, 12 and 14 d of differentiation. Expression was normalized to the control ribosomal gene RPL13. Three biological replicates were performed. Error bars denote ±standard deviation.

The following figure supplement is available for figure 7:

**Figure supplement 1.** Oct1 and Oct4 ChIPseq read density at example co-bound genes.

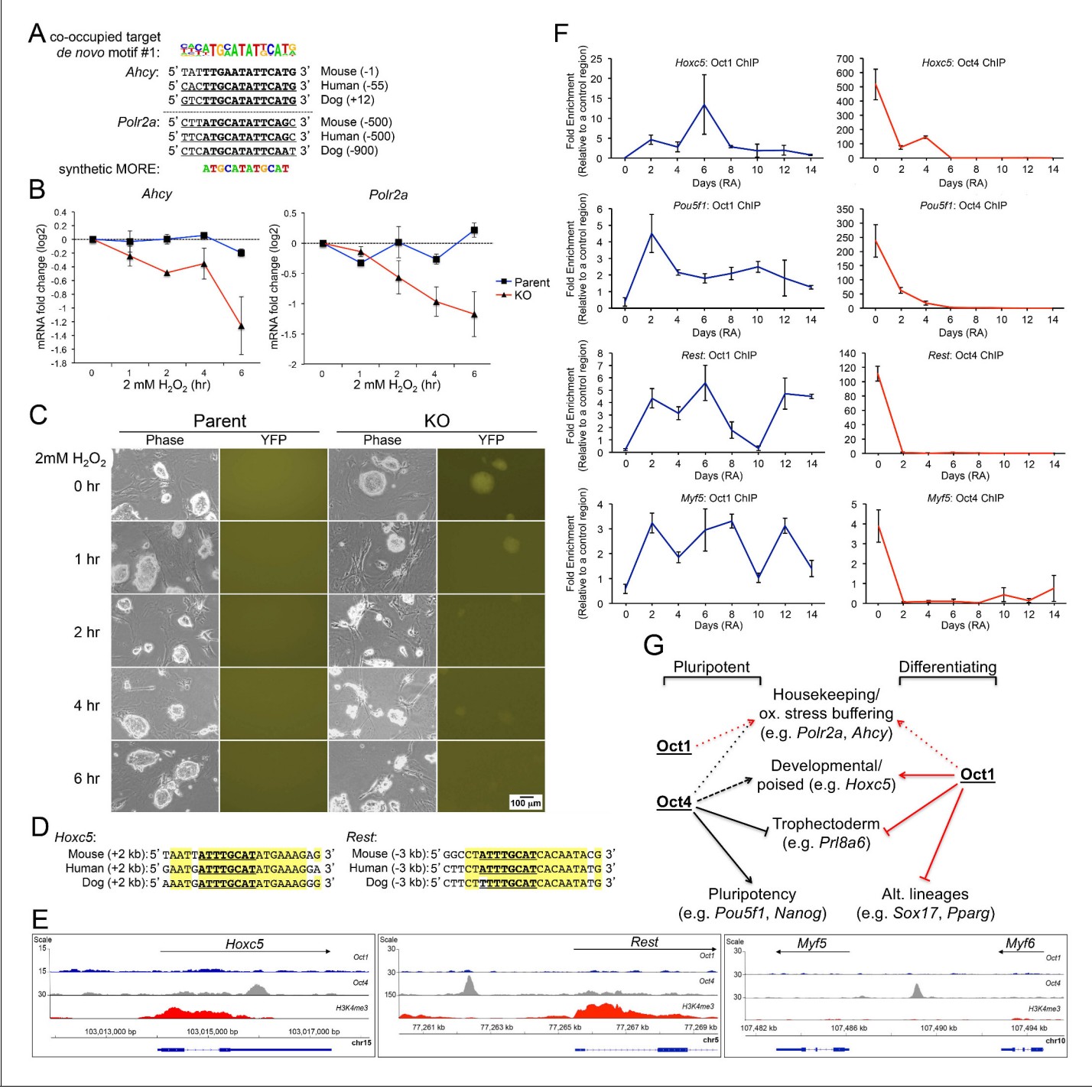

**Figure 8.** Oct1/Oct4 co-binding to MOREs in ESCs, and inducible Oct1 binding to poised targets upon differentiation. (**A**) Conserved MOREs at two genes (*Ahcy* and *Polr2a*) co-bound by Oct1 and Oct4 in ESCs. The top co-bound de novo motif from *Figure 7B* is shown at top. The MORE sequence (*Reményi et al., 2001*; *Tomilin et al., 2000*) is shown at bottom. Mammalian (mouse, human, dog) conservation is shown. MORE position relative to TSS is shown in parentheses. *Polr2a* contains two adjacent MOREs (*Kang et al., 2009*), only one of which is shown here. (**B**) Parent and derived KO ESCs were treated with 2 mM $H_2O_2$ for the indicated times. *Ahcy* and *Polr2a* $\log_2$ mRNA levels were assessed by RT-qPCR. Three biological replicates were performed. Error bars denote ±standard deviation. (**C**) Microscopic images of the same cells during the treatment timecourse. (**D**) Conservation of the octamer sequence in the *Hoxc5* 3' UTR and *Rest/Nrsf* upstream region. Mammalian (mouse, human, dog) conservation is shown. Octamer element position relative to TSS is shown in parentheses. (**E**) Genome tracks showing ChIPseq enrichment at *Hoxc5*, *Rest* and upstream of *Myf5*, for Oct1 (blue), Oct4 (gray) or H3K4me3 (red). (**F**) ChIP-qPCR differentiation timecourse of four targets that exclusively bind Oct4 in ESCs. Fold enrichments using Oct1 and Oct4 antibodies at *Hoxc5*, *Pou5f1*, *Rest* and *Myf5* are shown. (**G**) Model for Oct1 and Oct4 function in ESCs and their differentiated progeny. The

*Figure 8 continued on next page*

**Figure 8 continued**

example of neuronal differentiation is shown. In stem cells, Oct1 and Oct4 collaborate at constitutively expressed MORE-containing targets such as *Polr2a* and *Ahcy* to insulate them against oxidative stress (red and black short dashed lines). Oct4 poises developmental genes of all embryonic lineages (long dashed black line) and repress trophectoderm-specific genes (solid black block line). Oct4 additionally activates pluripotency genes (solid black arrow). In differentiating cells, Oct1 occupies MORE genes in response to oxidative stress and buffers their expression, as described previously (*Kang et al., 2009*; *Shakya et al., 2011*). Oct1 also contributes to eventual lineage-specific developmental gene activation (solid red line), and alternate developmental lineage gene repression (including trophectoderm, solid red block line).

## Cell culture

ESCs were cultured as previously (*Shakya et al., 2015a*) with 2i conditions: the ERK inhibitor PD0325901 (1 µM, LC Laboratories) and the GSK3 inhibitor CHIR99021 (3 µM, LC Laboratories). 4-OHT (Sigma) was dissolved in ethanol and used at 500 nM for 24 hr. Two methods were used to generate EBs. Low-attachment dishes were used to generate WT and Oct1-deficient EBs for microscopic analysis, RT-qPCR and the generation of neurons. Briefly, ESCs were trypsinized and feeders depleted by binding to gelatin-coated dishes for 30–60 min. ESC suspensions were plated on low-attachment dishes for 5–7 days. For cardiomyocyte differentiation, the hanging drop method (*Wang and Yang, 2008*) was used in order to generate single EBs in 96-well plates. Individual EBs were then used to generate cardiomyocyte colonies in 24-well plates. Generation of neurons was accomplished as in (*Bain et al., 1995*), with modifications. Briefly, EBs were formed for 4 days using low-attachment dishes, followed by culture for a further 4 days as EBs in 0.1 µM RA/DMEM. After 8 days, EBs were trypsinized and cultured for 8 days in 1:1 F12:DMEM, 10 µg/mL insulin (SAFC Biosciences), 5.5 µg/mL transferrin and 38.7 µM sodium selenite (ThermoFisher) on laminin/poly-L-lysine-coated ChamberSlides (Corning). Cells in *Figure 2E–F* were cultured for eight additional d. For $H_2O_2$ treatment, ESCs were seeded 24 hr prior to treatment on 6-well plates with sparse feeders. Cells were treated with 2 mM $H_2O_2$ (Sigma) for the indicated times.

## Immunoblotting

Antibodies for immunoblotting were as follows: Oct4, Santa Cruz sc-5279; Oct1, Bethyl A301-716A + A301–171A; Nanog, GeneTex GTX100863; Sox2, GeneTex GTX101507; GAPDH, EMD Millipore, MAB374; α-Tubulin, Santa Cruz sc-5286.

## RT-qPCR

RNA was isolated using TRIzol (Thermo Fisher, Waltham MA), followed by RNAeasy purification (Qiagen) using the RNA cleanup procedure. cDNA was synthesized using SuperScript III and random hexamers (Thermo Fisher). RT-qPCR oligonucleotide primers are listed in *Supplementary file 3*.

## Lentiviral Oct1 complementation

The *Oct1* (*Pou2f1*) cDNA and IRES (internal ribosomal entry site) elements were amplified and cloned together by overlap PCR. In the first PCR, primers to the 5' end of *Oct1* containing a *Not*I restriction site and to the 3' end of *Oct1* that contained a 5' extension of IRES-complementary DNA were used. The sequences were: Oct1-NotI-For: 5'-AATGAAAAAAGCGGCCGCCATGAATAATCCATCAGAAAC-3'; Oct1-Rev-IRES: 5'-TTAGGGGGGGGGGGAGGGATCTTCACTGTGCCTTGGAG-3'. In the second PCR, an IRES sequence was amplified using primers to the 5' end of the IRES containing a 5' extension of DNA complementary of the *Oct1* 3' end, and primers to the 3' end of the IRES containing an *Nde*I restriction site. The sequences were: IRES-overlap-FOR: 5' AGATCCCTCCCCCCCCCCTAACGTTACTGGCCGAA-3'; IRES-Rev-NdeI: 5'- GGGAATTCCATATGTGTGGCCATATTATCATCGTGT-3'. The third PCR used as a template the PCR products from the first two rounds, along with the Oct1-NotI-For and IRES-Rev-NdeI primers. This process generated a DNA fragment containing an *Oct1* cDNA fused to an IRES at the 3' end, along with a *Not*I site at the 5' terminus and an *Nde*I site at the 3' terminus. The fragment was cloned into the optimized, self-inactivating, nonreplicative pHAGE lentiviral vector using the *Not*I and *Nde*I restriction sites. To insert a *Puro* cassette after the IRES, the cDNA was amplified using primers containing 5' *Nde*I and 3' *Cla*I restriction sites. The sequences were Puro-NdeI-For: 5'- GGAATTCCATATGATGACCGAG

TACAAGCCCACGGT-3'; Puro-Clal-Rev: 5' GGTTTATCGATTCAGGCACCGGGCTTGC-3'. Because the IRES apparently attenuated expression of the *Puro* resistance cassette in this vector, puromycin selection was performed at 0.75 µg/mL. To generate an empty vector control, the vector was cut with *Nde*I and *Not*I, filled in with Klenow fragment, and re-ligated.

## Immunofluorescence

Immunofluorescence was performed as described previously (*Kang et al., 2013*), using mouse monoclonal antibodies against $\beta$-tubulin-III (R and D Systems MAB1195) and rabbit polyclonal antibodies against YFP (Life Technologies A6455). Secondary antibodies used were goat anti-rabbit-Alexa568 (Life Technologies A-11011) and goat anti-mouse-Alexa488 (Life Technologies A-11001).

## Teratoma formation

Teratomas were generated as described (*Nelakanti et al., 2015*) by injecting parent or KO ESCs into contralateral flanks of female NCr Nude mice (NCRNU-F, Taconic). Mice were sacrificed at four wk. Tumors were excised, washed with cold PBS and weighed. 1/3 of the excised tumor was used to make lysates for protein analysis using a Dounce homogenizer with RIPA lysis buffer (50 mM Tris pH 7.4, 150 mM NaCl, 0.1% SDS, 0.1% sodium deoxycholate, 1 mM EDTA and protease inhibitors [Roche]) on ice. Lysates were centrifuged 10,000 $\times$ *g* for 10 min. Supernatant protein concentrations were normalized using Bradford assays. 6$\times$ Laemmli sample buffer was added. The mixture was boiled for 5 min and resolved using a 10% SDS-PAGE gel. The remainder of the tumor was fixed in formaldehyde, paraffin-embedded, sectioned and H and E stained for histological analysis by a blinded pathologist.

## RNAseq

RNA was prepared from three independent cultures of undifferentiated or 14 d RA-differentiated parent *Pou2f1*$^{fl/fl}$ or 4-OHT treated *Pou2f1*$^{\Delta/\Delta}$ ESCs. Concentration was determined using a Quant-iT RNA assay kit and a Qubit fluorometer (Thermo Fisher). Intact poly(A) RNA was purified from total RNA samples (100–500 ng) with oligo(dT) magnetic beads, and stranded mRNA sequencing libraries were prepared as described using the Illumina TruSeq mRNA library preparation kit. Purified libraries were qualified on an Agilent Technologies 2200 TapeStation using a D1000 ScreenTape assay. Molarity of adapter-modified molecules was defined by qPCR using the Kapa Biosystems Library Quant Kit. Individual libraries were normalized to 10 nM and equal volumes were pooled in preparation for Illumina sequencing. Sequencing libraries (25 pM) were chemically denatured and applied to an Illumina HiSeq v4 paired end flow cell using an Illumina cBot. Hybridized molecules were clonally amplified and annealed to sequencing primers with reagents from an Illumina HiSeq PE Cluster Kit v4-cBot. Following transfer of the flowcell to an Illumina HiSeq 2500 instrument (HCS v2.2.38 and RTA v1.18.61), a 125-cycle paired-end sequence run was performed using HiSeq SBS Kit v4 sequencing reagents. Fastq data quality were checked using Fastqc version 0.10.1 (http://www.bioinformatics.babraham.ac.uk/projects/fastqc/). Quality scores dipped towards the 3' end of the reads, so reads were trimmed at 50 bases to eliminate poor-quality data. The resulting 50-base reads were aligned to the mouse mm10 genome (GRCm38, December 2011) plus splice junctions using novoalign version 2.08.01 (http://www.novocraft.com). Alignments to splice junctions were translated back to genome coordinates using the SamTranscriptomeParser application in the USeq package (*Nix et al., 2010*). Aligned reads were quality checked using the Picard tools' CollectRnaSeqMetrics command (https://broadinstitute.github.io/picard/). On average 99.0% of the reads aligned to the mouse genome, with 78% of reads providing unique alignments, and 86% of reads providing alignments to protein coding and UTR regions of the genome. Tests for differential gene expression were performed with DESeq2, version 1.10.0 (*Love et al., 2014*). Genes with a count of at least 50 in one or more samples were tested. Genes showing at least 2.5-fold change of expression at an adjusted *p*-value of <0.01 were selected as differentially expressed. Figures were generated in R version 3.2.3 (http://www.r-project.org) using functions from the gdata and gplots libraries.

## ChIP/ChIPseq

ChIP was performed as described (*Shakya et al., 2015a*). ChIP oligonucleotide primers are listed in *Supplementary file 3*. Antibodies used were the following: Oct1 (Bethyl, a mixture of A301-

716A + A301–717A), Oct4 (Santa Cruz, sc-8629) and H3K4me3 (Millipore, 07–473). ChIPseq was performed as described previously (*Shakya et al., 2015a*, *2015b*), using a single IP per condition and clones of parent or derived Oct1-deficient ESCs. For ChIPseq, reads were aligned to the mouse reference genome (mm10) with the Burrows-Wheeler Aligner (BWA, version: 0.5.9). Reads were filtered for alignment quality of >Q10 and duplicates were removed using Picard tools (function MarkDuplicates). After filtering there were 21.1 (H3K4me3), 27.3 (Oct1), and 23.7 (Oct4) million reads. MACSv2 peak caller (version: 2.1) was used to call ChIPseq regions of enrichment with the following parameters (-p 1e-5 –nomodel –shiftsize <fragment_length/2> for Oct1, Oct4 and -p 1e-2 –broad for H3K4me3). To estimate the –shiftsize parameter (predominant fragment length divided by 2) we performed strand cross-correlation analysis using SPP R package (version: 1.10.1) with default parameters. Peaks overlapping with ENCODE blacklisted regions were filtered using BEDtools (function itersectBed). We also discarded peaks localized to mitochondria, chromosome Y, and unmapped contigs. After filtering we had 692 (Oct1), and 8673 (Oct4) peaks. Signal to noise ratio was assessed by calculating normalized strand coefficient (NSC) and relative strand correlation (RSC) using the SPP R package with default parameters (version: 1.10.1). The obtained values of NSC and RSC (H3K4me3: 2.28, 1.25; Oct1: 1.02, 1.45; Oct4: 1.05, 2.32) indicate highly enriched datasets with large fragment-length peak as compared to read-length peak. The NSC value for Oct1 transcription factor was somewhat smaller but typical for high quality datasets generated for factors with small numbers of genuine binding sites (692 MACS2-identified peaks for Oct1). We used MACSv2 function bdgdiff to build fold-enrichment signal tracks for all positions in the genome. Signal tracks were converted to TDF files using igvtools (https://www.broadinstitute.org/igv/igvtools). Peaks were allocated to genes using the annotatePeaks.pl program from HOMER suite (Hypergeometric Optimization of Motif Enrichment, version: 4.7, http://homer.salk.edu/homer/) by determining the closest RefSeq transcription start sites of the genes to the peaks. Functional enrichment analysis was performed using the findGO.pl program from HOMER and Bonferroni as well as Benjamini and Hochberg correction for multiple testing corrections. Robustness of the analysis was confirmed using MEME-ChIP (*Machanick and Bailey, 2011*), which generate highly similar motifs.

## Motif analysis

Transcription factor enrichment within ChIPSeq peaks (de novo motif discovery and known motif matching) was determined using findMotifsGenome.pl program from HOMER. Motif analysis was run on overlapped and separately on unique Oct1 and Oct4 ChIPseq peaks. Oct1 and Oct4 ChIPseq peak overlaps were defined by requiring the distance between peak summits to be ≤100 bp. Motif lengths of 6–24 bp were identified within 200 bp regions centered on peak summits and an option of random background was selected for motif discovery.

## Acknowledgements

We thank Drs. Time Formosa and David Stillman for critical reading of the manuscript. We thank Susan Tamowski and the University of Utah Health Sciences Center Transgenic and Gene Targeting Core for assistance generating Oct1 deficient and conditional-inducible ESCs, and for assistance with blastocyst injections to generate chimeras. We thank James Cox and the University of Utah Metabolomics Core. We thank Brian Dalley and the University of Utah High-Throughput Genomics Core, Brett Milash and the University of Utah Bioinformatics Core, and the University of Utah Center for High Performance Computing for assistance with RNAseq. This work was supported by grants from the University of Utah Department of Pathology, NIH/NIAID (R01AI100873) and an endowment (Watkins Endowed Chair) to DT

## Additional information

### Competing interests

AR: Senior editor, *eLife*. The other authors declare that no competing interests exist.

## Funding

| Funder | Grant reference number | Author |
| --- | --- | --- |
| National Institute of Allergy and Infectious Diseases | R01AI100873 | Dean Tantin |

The funders had no role in study design, data collection and interpretation, or the decision to submit the work for publication.

## Author contributions

ZS, JK, AS, MT, Data curation, Formal analysis, Investigation, Writing—original draft, Writing—review and editing; EAJ, Formal analysis, Writing—original draft, Writing—review and editing; AR, Supervision, Funding acquisition, Writing—original draft, Project administration, Writing—review and editing; DT, Conceptualization, Supervision, Funding acquisition, Writing—original draft, Project administration, Writing—review and editing

## Author ORCIDs

Dean Tantin, http://orcid.org/0000-0003-1354-8385

## Ethics

Animal experimentation: This study was performed in strict accordance with the recommendations in the Guide for the Care and Use of Laboratory Animals of the National Institutes of Health. All of the animals were handled according to approved institutional animal care and use committee (IACUC) protocols (#14-06015) of the University of Utah. Every effort was made to minimize suffering.

## Additional files

### Supplementary files

• Supplementary file 1. Normalized RNAseq gene expression changes.

• Supplementary file 2. Genes identified by ChIPseq in comparison to RNAseq data.

• Supplementary file 3. Oligonucleotides for RT-qPCR and ChIP.

### Major datasets

The following dataset was generated:

| Author(s) | Year | Dataset title | Dataset URL | Database, license, and accessibility information |
| --- | --- | --- | --- | --- |
| Shen Z, Kang J, Shakya A, Tabaka M, Jarboe EA, Regev A, Tantin D | 2017 | Enforcement of developmental lineage specificity by transcription factor Oct1 | https://www.ncbi.nlm.nih.gov/geo/query/acc.cgi?acc=GSE85063 | Publicly available at the NCBI Gene Expression Omnibus (accession no: GSE85063) |

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
