## [Decision Letter]

Thank you for submitting your article "Enforcement of developmental lineage specificity by transcription factor Oct1" for consideration by *eLife*. Your article has been reviewed by three peer reviewers, and the evaluation has been overseen by a Reviewing Editor and Fiona Watt as the Senior Editor. The reviewers have opted to remain anonymous.

The reviewers have discussed the reviews with one another and the Reviewing Editor has drafted this decision to help you prepare a revised submission.

Summary:

In this manuscript entitled: "Enforcement of developmental lineage specificity by transcription factor Oct1", Shen et al. have addressed the role of Oct1 in pluripotency and differentiation, by analysing the phenotype of Oct1 null ES cells via both conditional deletion in vitro and after deriving a null ES line. They show that the Oct1 null cells can be propagated under pluripotency conditions with no overt phenotypic differences from the parental line, but cannot undergo normal differentiation when challenged using multiple assays (differentiation to cardiomyocytes and neurons in vitro, and capacity to form teratomas and contribute to chimeras in vivo). Consistent with this, while pluripotency genes are down-regulated normally in the differentiation protocols, differentiation markers are expressed aberrantly. The authors then use RNAseq and ChIPseq (for Oct1, Oct4 and H3K4me3) to address the role of Oct1 during RA-induced ES differentiation. From this, they conclude that part of its effect is to insulate a set of target genes, co-bound by Oct4, from repressive effects of oxidative stress. They also reveal a dynamic interplay between Oct4 and Oct1 binding, where Oct1 takes over from Oct4 as Oct4 levels drop, and suggest this is required to promote lineage-appropriate and suppress lineage-inappropriate expression during differentiation. Overall, the findings are novel and interesting, and the manuscript is well presented. The reviewers have requested the following revisions, which should be addressed in the revised manuscript.

Essential revisions:

1) We request that the authors distinguish between two possible explanations of their data, as detailed below: Oct1/Pou2f1 belongs to the POU family, in which the members share the POU domain as a DNA binding domain. Oct4/Pou5f1 is also a member of the POU family and known as a key transcription factor for establishment and maintenance of pluripotency. The authors have shown that Oct1 and Oct4 bind to mostly different targets in ES cells, although a small proportion of their target sites are shared, indicating their distinct role. Most of the genes occupied by Oct1 do not show differential expression during differentiation, suggesting that these are housekeeping genes. If the regulation of these housekeeping genes by Oct1is required for the survival of a set of differentiated cells, which may happen depending on particular metabolic states, the deficiency of Oct1 would eliminate these cells. Although the authors emphasized that the dysregulation of developmentally-regulated genes is responsible for the defect of differentiation of Oct1-null ES cells, it could alternatively be due to the elimination of the differentiated cells rather than the defect of differentiation event. RA-induced ES cell differentiation is quite a crude system for addressing this point. The authors are encouraged to apply a cleaner system of differentiation and to induce the conditional KO at intermediate states of differentiation event to distinguish these two possibilities.

2) Related to this, the authors are requested to show whether the defects in expression of differentiation marker genes and misexpression of alternative marks shown in Figure 1 and Figure 2 occurs equally across the whole population or only in a subset of cells. These are both possible interpretations of expression data taken from the whole population, but would could lead to different interpretations of the function and mechanism of Oct1 in controlling cell fate. The authors are also requested to clarify whether the defect in the formation of β-tubulin III-expressing neurons in the null versus wild-type cells shown in Figure 2 corresponds to a failure to fully differentiate, or if the null cells are showing slower differentiation kinetics. Was this experiment continued beyond the normal time period of the differentiation protocol, and if so, did the null cells eventually 'catch up' or did they genuinely fail to robustly produce the appropriate neurons?

3) It was previously shown that Oct1-knockdown (~ 50%) in ESCs does not result in any phenotype during neuronal differentiation because of functional redundancy with Oct2 (Theodorou et al. Genes & Dev. 2009. 23: 575-588). To address this inconsistency with their data, the authors are requested to carry out rescue experiments to make sure that Oct1 can rescue the differentiation capacity of Oct1 KO ES cells (e.g. with a constitutively-active Oct1 transgene), in order to confirm the Oct1-dependency of the phenotype.

4) The authors are also requested to deal with the following queries related to data presentation or analysis:i) The authors have generated two germline Oct1 KO lines and littermate WT lines as well as two Oct1 inducible KO lines, however it is not clear whether they have carried out all their qPCR and RNA seq analysis from both lines and pooled together or they have just used one line. It is important to carry out all their experiments in both lines and plot them separately so one can see the variation between the different lines (same genotype) and that between the WT and KO.

ii) The authors have not indicated how many biological replicates they have carried out for their ChIP-seq data. They are requested to clearly indicate the variation between the replicate libraries, which can then be pooled for later analysis.

iii) The authors have only used one algorithm to carry out motif analysis, this analysis should be repeated using a different algorithm (such MEME-ChIP).

iv). Similarly, the authors should examine whether Oct2 can functionally compensate the loss Oct1 and rescue the differentiation phenotype of Oct1 KO ES cells.

[Editors' note: further revisions were requested prior to acceptance, as described below.]

Thank you for resubmitting your work entitled "Enforcement of developmental lineage specificity by transcription factor Oct1" for further consideration at *eLife*. Your revised article has been favorably evaluated by Fiona Watt (Senior Editor), a Reviewing Editor, and two reviewers.

The manuscript has been improved but there are some remaining issues that need to be addressed before acceptance, as outlined below:

As you will see from the reviewers’ comments, there was a remaining concern regarding the interpretation of some of your data. In particular, the data presented in Figure 4 do not unequivocally demonstrate that switching from Oct4 to Oct1 is required for differentiation (as noted in the comments from reviewer 1). Therefore, we ask that you add Oct1 staining to the YFP staining presented and, in addition, that you address in the text the possible alternative explanations for these results (you might also consider changing the title in this light). If you do not feel able to make these changes, we would require the rescue experiment outlined by reviewer 1 to be included before the manuscript could be accepted for publication. We also request that you attend to both of the issues raised by reviewer 3, which can be dealt with by textual changes.

*Reviewer #1:*

In the revised manuscript, the authors made several changes to address the points mentioned by the reviewers. The most important point the authors should confirm was the functional confirmation of the switching from Oct4 to Oct1 on induction of differentiation-associated genes to undergo proper differentiation event. Neuronal differentiation was well-established system and the combination of the inducible KO of Oct1 would allow them to give a clear answer to this question as the reviewers expected. However, the experiment performed by the authors was not well organized and the result was equivocal. If the switch from Oct4 to Oct1 at the beginning of differentiation event is functionally important, the constitutive KO ES cells will fail to give rise to terminally differentiated cells whereas the conditional KO induced after switching will differentiate, even in lower efficiency than WT ES cells. However, induction of KO event at day 4 of differentiation culture, the time point when Oct1 was already replace Oct4 at several target sites as shown in Figure 8, completely abolished differentiation to mature neurons. This result suggested that Oct1 function is required for the late period of differentiation event or maintenance of the mature differentiated phenotype in cell-type-dependent manner. The assessment of KO event at cellular level was incomplete in this experiment. Why did the authors detect YFP expression, an indirect marker of inducible KO, rather than the loss of Oct1 protein with anti-Oct1 Ab?

An alternative way to address this point precisely is the rescue by the inducible Oct1 expression in Oct1-null ES cells after induction of differentiation. According to the answers to the reviewers' comments, the authors tried to rescue Oct1-null ES cells by transfection of constitutively-active Oct1 transgene at different time point, and failed. Why not using the inducible expression system such as Tet-on system to drive Oct1 transgene? It is well-established system and commercially available.

The strict confirmation of this point is very important to support the hypothesis the authors proposed. Without such confirmation, the other data is insufficient to support the hypothesis. The role of Oct1 for normal proliferation, protection against stress, and maintenance of certain metabolic state in differentiated cell types would explain the defect of Oct1-null ES cells in chimera assay, teratoma assay and slower growth of EBs rather than the defect in differentiation event.

*Reviewer #3:*

I think Tantin and colleagues have addressed most of the major concerns raised by the reviewers in the first round. After fixing few minor issues (see below), I would recommend the publication of this manuscript in *eLife*.

Issue 1: Figure 6. The authors compare RNA-seq data of parental vs. KO in both undifferentiated and differentiated conditions. The authors conclude that the KO show more gene expression differences during differentiation. However, it is important to show variability across parental and KO samples separately in both conditions. i.e. one would expect that cells are more heterogeneous during RA-differentiation compared to pluripotency state. So, it is expected to see more difference in RA-differentiation state.

Issue 2: Figure 7 (bottom). the known Oct4-Sox2 motif was enriched in 22.9% of Oct4 unique peaks as compared to Oct4 motif which was enriched 15.3%. This does not make sense as Oct4 motif is the exact first half of the Oct4-Sox2 composite motif. One would expect Oct4 motif to be present in all Oct4-Sox2 (22.9%) sites plus few more sites that only contain Oct4 motif. The authors need to clarify what those percentages mean in the legend.

---

## [Author Response]

*Essential revisions:*

*1) We request that the authors distinguish between two possible explanations of their data, as detailed below: Oct1/Pou2f1 belongs to the POU family, in which the members share the POU domain as a DNA binding domain. Oct4/Pou5f1 is also a member of the POU family and known as a key transcription factor for establishment and maintenance of pluripotency. The authors have shown that Oct1 and Oct4 bind to mostly different targets in ES cells, although a small proportion of their target sites are shared, indicating their distinct role. Most of the genes occupied by Oct1 do not show differential expression during differentiation, suggesting that these are housekeeping genes. If the regulation of these housekeeping genes by Oct1is required for the survival of a set of differentiated cells, which may happen depending on particular metabolic states, the deficiency of Oct1 would eliminate these cells. Although the authors emphasized that the dysregulation of developmentally-regulated genes is responsible for the defect of differentiation of Oct1-null ES cells, it could alternatively be due to the elimination of the differentiated cells rather than the defect of differentiation event. RA-induced ES cell differentiation is quite a crude system for addressing this point. The authors are encouraged to apply a cleaner system of differentiation and to induce the conditional KO at intermediate states of differentiation event to distinguish these two possibilities.*

The short answer to this point is that we performed the experiment outlined by the reviewers (Figure 4), who indicated that we differentiate inducible-conditional ESCs using a precise system (we chose neurons), exposing cells to tamoxifen at different time points. Neurons are generated by first differentiating ESCs into EBs in low-attachment plates, followed by subsequent plating in CultureSlide chambers with insulin, selenium and transferrin. Cells then grow out from the fragmented EBs and fill the chamber as a monolayer. Deletion of Oct1 by tamoxifen administration is inefficient (Figure 4). Only a fraction of ESCs within colonies become YFP^+^, and not all of those delete both Oct1 alleles. Unlike in ESCs, where we can pick and characterize individual colonies, in differentiating neurons we do not know which yellow cells have deleted one vs. both Oct1 alleles, and so we have to rely on the power of numbers. We treated differentiating cells with 4-OHT at different points during EB formation and subsequent differentiation of neurons. Deletion in EBs was highly inefficient. At only one timepoint (4 days after plating in chambers, 12 days into differentiation) did we generate YFP-positivity at an efficiency (40-50%) that allowed us to conduct this experiment. We fixed the neurons at endpoint and performed immunofluorescence with antibodies against YFP (red) and β-III-tubulin (green). We observe a much smaller fraction of YFP^+^ cells that express β-III-tubulin, and an even smaller fraction of YFP^+^ cells that form true arborized neurons (2 out of ~750 cells analyzed). These few cells are likely to be Oct1 heterozygous, though it is not possible to prove this. Representative images and quantified data from these results are now shown in Figure 4. For completeness, we also now show the effect of conditional Oct1 deletion in ESCs on neuron formation. The effect is identical to that seen in germline-deficient ESCs, as expected (Figure 4).

Regarding the larger reviewer rationale behind this experiment (that cells might be dying and skewing populations): we and others have shown that although it binds and regulates housekeeping genes Oct1 is not a housekeeping factor. Oct1 is dispensable for cell viability. Oct1 loss, either by knockout or RNAi) does not impede proliferation rates in standard culture conditions such as we use here. Published examples include primary MEFs, A549 and MD-MBA-231 cell lines, fetal liver blood progenitors, developing and activated primary T cells, total B cells and macrophages. Submitted work using CRISPR shows the same result in primary MEFs and MCF-7 cells. In unpublished work, deletion in the gut retains normal architecture and homeostasis, maintaining deletion for >100 days. Instead Oct1 loss specifically impairs regeneration following epithelial cell damage. The ratio of oxidative to glycolytic metabolism in these situations is changed as expected, but in rich media and without some form of stress, cells are surprisingly resilient to such metabolic alterations. In our EB, cardiomyocyte, true neuron and RA/neuronal differentiation experiments, we note no differences in cell death (e.g., Supp. Figure 1—figure supplement 1, Figure 2—figure supplement 2 and Figure 4—figure supplement 2). Finally, work from others shows that germline Oct1 null embryos, in which lethality due to trophoblast defects is circumvented with tetraploid complementation, do not die until E9.5, which seems incompatible with a model that has large populations of embryonic cells dying upon implantation and initial specification/differentiation.

*2) Related to this, the authors are requested to show whether the defects in expression of differentiation marker genes and misexpression of alternative marks shown in Figure 1 and Figure 2 occurs equally across the whole population or only in a subset of cells. These are both possible interpretations of expression data taken from the whole population, but would could lead to different interpretations of the function and mechanism of Oct1 in controlling cell fate. The authors are also requested to clarify whether the defect in the formation of β-tubulin III-expressing neurons in the null versus wild-type cells shown in Figure 2 corresponds to a failure to fully differentiate, or if the null cells are showing slower differentiation kinetics. Was this experiment continued beyond the normal time period of the differentiation protocol, and if so, did the null cells eventually 'catch up' or did they genuinely fail to robustly produce the appropriate neurons?*

There are two points here, both of which are valid. The first is whether the defects observed with Oct1 loss are partial in nature across many cells, or whether a much smaller fraction of cells differentiate completely normally. The data in Figure 2 already provides a sense of this: in the neuron differentiation cultures we can find multiple examples of Oct1 deficient cells with poor β-III-tubulin staining, but nevertheless show staining above background. Rarely however we do see positive cells that have a neuronal appearance (as quantified in panels 2D, 4H and 4J, though 4J may be quantifying heterozygotes). To illustrate this better, we now show new images with quantifications (Figure 2). Therefore, in the absence of Oct1 we observe many more Oct1 deficient cells with poor/incomplete differentiation, at least based on β-III-tubulin staining and arborization as criteria. Regarding the second, kinetic, question, the data in Figure 2—figure supplement 1 shows that even early during differentiation into EBs, abnormal β-III-tubulin staining is evident in the differentiating knockout cells. To further address this comment, we fixed and stained germline Oct1-deficient ESCs that had been differentiated into neurons for longer periods. We again used the terminal differentiation marker β-III-tubulin. At no point do Oct1 deficient cells “catch up”, indicating that the defects observed with Oct1 deficiency are not due to changes in kinetics. Data from 8 day-extended cultures are shown in Figure 2. We reference the 16 day-extended culture results as data not shown.

*3) It was previously shown that Oct1-knockdown (~ 50%) in ESCs does not result in any phenotype during neuronal differentiation because of functional redundancy with Oct2 (Theodorou et al. Genes & Dev. 2009. 23: 575-588). To address this inconsistency with their data, the authors are requested to carry out rescue experiments to make sure that Oct1 can rescue the differentiation capacity of Oct1 KO ES cells (e.g. with a constitutively-active Oct1 transgene), in order to confirm the Oct1-dependency of the phenotype.*

We added language to the Discussion section acknowledging the above study, which shows that partial Oct1 knockdown reduces neuron formation in the context of an Oct2 knockout. It is not clear there is any inconsistency here, given that the knockdown in the prior study was only 50%, and given that transient transfection of siRNAs was used. Oct1 germline null heterozygotes are viable and fertile, suggesting that a 50% reduction is insufficient for the phenotypes we describe here to manifest. Additionally, we use multiple cell lines derived using two different methods (germline Oct1-deficient ESCs and WT lines derived in parallel, and conditional-inducible knockout ESCs in which Oct1 is deleted and tested compared to matched vehicle-treated parental controls). Both methods arrive at the exact same conclusions. We did attempt additional experiments to address this point. The language is not clear but generating a mouse transgene was not feasible for this study. As an alternative, we used the RA differentiation paradigm with MSCV viral transduction of a vector construct encoding Oct1 and a puromycin resistance cassette. We attempted viral transductions at nearly every day of RA differentiation. We attempted multiple infections of concentrated virus. Nonetheless infection rates were <10%, possibly because there is not much cell division at any given time point, and the results were uninterpretable. It would be possible to circumvent this issue by sub-cloning the mouse Oct1 cDNA into a lentiviral vector, however even this approach has caveats. For example, Oct1 levels fluctuate during differentiation (Figure 2) and it is not clear if constitutive expression from a viral promoter constitutes “rescue.” The relative levels of Oct1 and Oct4 clearly have effects on gene occupancy, and it is likely that similar mass-action effects occur with Oct1, Oct2, Oct6 and other neuronal Oct binding transcription factors. It is also not clear whether the cells remaining after infection and selection are similar in composition to un-manipulated RA-differentiated cells, although empty vector controls partially address this. Lastly, the timing of the infection may be important, so one could try to complement Oct1 at day 2, 4, 6, 8, 10, 12 or 14. For all these reasons it would take many additional months to carry through with such an experiment, with a high potential for uninterpretable results.

*4) The authors are also requested to deal with the following queries related to data presentation or analysis:i) The authors have generated two germline Oct1 KO lines and littermate WT lines as well as two Oct1 inducible KO lines, however it is not clear whether they have carried out all their qPCR and RNA seq analysis from both lines and pooled together or they have just used one line. It is important to carry out all their experiments in both lines and plot them separately so one can see the variation between the different lines (same genotype) and that between the WT and KO.*

We did not mix lines, but did confirm the phenotypes shown with multiple lines. We have clarified in the manuscript text that data are shown for a single example of either germline or conditional KO and controls, rather than pooling different lines. We anticipated this concern in the manuscript, which is why we also generated inducible-conditional ESCs and tested knockout in these lines compared to the vehicle-treated parent line.

*ii) The authors have not indicated how many biological replicates they have carried out for their ChIP-seq data. They are requested to clearly indicate the variation between the replicate libraries, which can then be pooled for later analysis.*

We have clarified the text that ChIPseq libraries were prepared exactly as in Shakya et al. MCB 2015 and Shakya et a JEM 2015, using a single a IP per condition, from single clones of parent or matched derived Oct1 deficient ESCs. The ChIP material was first validated using *Pou5f1* for Oct4 and *Polr2a* for Oct1. H3K4me3 was used as an internal quality control. All ChIP-qPCR in the manuscript uses at least three experimental replicates, and these results substantiate the ChIPseq. Analysis of the DNA sequences within the ChIPseq peaks robustly identifies octamer DNA binding sites for both Oct1 and Oct4, and specific octamer variants for sites co-bound by both proteins (below). We recognize that this is not ideal, but because a large number of reads is required in order to validate each experimental condition, splitting the cells three ways, then performing ChIP and sequencing separately, then combining again, increases expense without a commensurate increase in the likelihood of generating meaningful data.

*iii) The authors have only used one algorithm to carry out motif analysis, this analysis should be repeated using a different algorithm (such MEME-ChIP).*

We performed the same analysis using MEME-ChIP. The top motifs found by both software packages are nearly identical, with similarly strong E-values for MEME-ChIP (HOMER uses P-values). This result confirms that our ChIPseq and analysis were robust (see the point above). We now state this in the manuscript text, though did not think the results were relevant enough to be included (they can be derived from our data and we can provide them on request). Of import here, both algorithms identify MOREs as the top sequence simultaneously occupied by Oct1 and Oct4. MOREs are known to bind Oct protein dimers in a novel configuration, and we had previously identified inducible Oct1 occupancy of MOREs following oxidative stress exposure in differentiated cells (PMID: 19171782). In the future, we hope to determine how Oct1 and Oct4 constitutively occupy sites that are inducibly occupied by Oct1 following stress in cells lacking Oct4.

*iv) Similarly, the authors should examine whether Oct2 can functionally compensate the loss Oct1 and rescue the differentiation phenotype of Oct1 KO ES cells.*

We have not studied Oct2 (and Oct6, and Oct11), both in order to make the manuscript straightforward and due to financial constraints. Whether or not Oct2 complements certain gene expression phenotypes is an interesting question, but regardless of the outcome would not alter any of the fundamental conclusions in the current study. We therefore maintain that such experiments are more appropriate for a future study.

[Editors' note: further revisions were requested prior to acceptance, as described below.]

*The manuscript has been improved but there are some remaining issues that need to be addressed before acceptance, as outlined below:*

*As you will see from the reviewers’ comments, there was a remaining concern regarding the interpretation of some of your data. In particular, the data presented in Figure 4 do not unequivocally demonstrate that switching from Oct4 to Oct1 is required for differentiation (as noted in the comments from reviewer 1). Therefore, we ask that you add Oct1 staining to the YFP staining presented and, in addition, that you address in the text the possible alternative explanations for these results (you might also consider changing the title in this light). If you do not feel able to make these changes, we would require the rescue experiment outlined by reviewer 1 to be included before the manuscript could be accepted for publication. We also request that you attend to both of the issues raised by reviewer 3, which can be dealt with by textual changes.*

The short answer to this is that we performed the rescue experiment, for which we show results with *Hoxa5* (new Figure 2). For technical reasons Oct1 and β-III-tubulin co-staining cannot be performed and generate satisfying results. This is why we used YFP, and one of the reasons why we engineered the LSL-YFP reporter cassette into the ESCs. The long explanation: Our results in the first resubmission show that tamoxifen treatment and Cre induction significantly diminishes formation of true neurons (Figure 4). We can interpret this as Oct1 deletion because of the robustness of the effects. If we had not observed strong differences, then the need to corroborate the YFP induction as reading out Oct1 deletion would have been more important. Nevertheless, we tried and failed to perform Oct1/β-III-tubulin co-staining, as outlined below. Because we knew that it would be a lingering issue, we then sub-cloned Oct1 into a lentiviral vector with a puromycin resistance cassette and infected differentiating Oct1-deficient ESCs. We chose to use RA-mediated differentiation, both because we have a clearer understanding of gene expression, and because at ~3 weeks these experiments take half as long compared to those using formation of true neurons. After several rounds of experiments, we now know that infection and selection with empty vectors at most differentiation timepoints, particularly late timepoints, changes expression of developmental targets genes such as *Hoxa5* and *Cdx2*. This confirms our initial concern, raised in the prior round of review, that selection skews the populations of cells remaining on the plate. However empty vector infection on day 4-5 (we performed two rounds of infection), generated minimal gene expression changes, at least on an averaged population level. This suggests, but does not prove, that population skewing by lentiviral infection is minimal at this time point. Subsequent experiments using vectors containing Oct1 shows that infection and selection infection significantly augments *Hoxa5* expression, complementing Oct1 loss. We hope these findings now address the reviewer’s remaining issues.

We are willing to change the title should it be deemed necessary. One possibility, more observational and less interpretive, would be “Transcription factor Oct1 both ensures robust expression of lineage- appropriate developmental genes, and represses expression of lineage-inappropriate genes, in differentiating ESCs.”

*Reviewer #1:*

*In the revised manuscript, the authors made several changes to address the points mentioned by the reviewers. The most important point the authors should confirm was the functional confirmation of the switching from Oct4 to Oct1 on induction of differentiation-associated genes to undergo proper differentiation event. Neuronal differentiation was well-established system and the combination of the inducible KO of Oct1 would allow them to give a clear answer to this question as the reviewers expected. However, the experiment performed by the authors was not well organized and the result was equivocal. If the switch from Oct4 to Oct1 at the beginning of differentiation event is functionally important, the constitutive KO ES cells will fail to give rise to terminally differentiated cells whereas the conditional KO induced after switching will differentiate, even in lower efficiency than WT ES cells. However, induction of KO event at day 4 of differentiation culture, the time point when Oct1 was already replace Oct4 at several target sites as shown in Figure 8, completely abolished differentiation to mature neurons. This result suggested that Oct1 function is required for the late period of differentiation event or maintenance of the mature differentiated phenotype in cell-type-dependent manner. The assessment of KO event at cellular level was incomplete in this experiment. Why did the authors detect YFP expression, an indirect marker of inducible KO, rather than the loss of Oct1 protein with anti-Oct1 Ab?*

*An alternative way to address this point precisely is the rescue by the inducible Oct1 expression in Oct1-null ES cells after induction of differentiation. According to the answers to the reviewers' comments, the authors tried to rescue Oct1-null ES cells by transfection of constitutively-active Oct1 transgene at different time point, and failed. Why not using the inducible expression system such as Tet-on system to drive Oct1 transgene? It is well-established system and commercially available.*

*The strict confirmation of this point is very important to support the hypothesis the authors proposed. Without such confirmation, the other data is insufficient to support the hypothesis. The role of Oct1 for normal proliferation, protection against stress, and maintenance of certain metabolic state in differentiated cell types would explain the defect of Oct1-null ES cells in chimera assay, teratoma assay and slower growth of EBs rather than the defect in differentiation event.*

There is no reason why Oct1 cannot both replace Oct4 at certain genes early in differentiation, and also continue to be required other aspects and fate transitions later in development. The reviewer is equating the ChIP timecourse result using RA differentiation culture in Figure 8 with 4-OHT treatment after 8 days of EB formation, followed by 4 days culture in chamber slides with neuron differentiation medium in Figure 4. Binding of Oct1 to target genes in these two assays systems can’t be equated.

We used 4-OHT at the only point at which deletion was efficient. The reason we used YFP instead of Oct1 for immunofluorescence is 1) the YFP antibody is superior to the Oct1 antibody for IF, 2) the Oct1 and β-III-tubulin antibodies are both rabbit polyclonal (we circumvented this by purchasing additional antibodies), 3) the staining protocols for Oct1 and β-III-tubulin are incompatible, relying on formaldehyde vs. methanol fixation, and no mutually compatible protocol could be found. But most importantly, we obtained the positive result that Oct1 inducible-conditional cells exposed to Cre recombinase 12 days after differentiation (4 days in chamber slides) fail to form neurons efficiently (Figure 4). The YFP staining would have been a significant issue of there had been no or small differences, in which case the possibility of heterozygous deletion and remaining Oct1 protein could be raised and a follow-up Oct1 stain would have been critical.

To further address this reviewer, we sub-cloned Oct1 into a lentiviral vector, providing us the opportunity to infect post-mitotic cells. As outlined in the response to the first round of reviewer comments, there are multiple factors that make this experiment non-ideal. Two are the fact that an infected and selected population of cells may not recapitulate the original population, due to preferences in the cell types or growth states infected, and the fact that Oct1 levels modulate significantly during differentiation and expression from a viral LTR may not recapitulate this complex regulation. The latter becomes a significant issue in light of the findings from the manuscript that relative levels of Oct1 and Oct4 (and presumably other Oct proteins) is a significant controller of gene occupancy and regulation. We infected RA-differentiating cultures of germline Oct1-deficient ESCs with lentiviruses encoding Oct1 (or empty vector) at multiple timepoints during differentiation, selected with puromycin, and tested gene expression by RT-qPCR. At most of these times, especially late in differentiation, infection and selection with empty vector was sufficient to skew expression of test genes such as *Hoxa5* and *Cdx2* (now shown). This result strongly suggests that infection and selection at these time points skews the population of cells under study. We therefore focused on one time at which gene expression changes caused by empty vector infection were minimal: day 4 and 5 of differentiation. We infect the cells twice to maximize efficiency, and apply puromycin selection 24 hr after the second infection throughout the remainder of the 14-day timecourse. After differentiation we collected RNA, prepared cDNA and tested gene expression. We confirmed that Oct1 was over- expressed in 293T cells transfected with Oct1-encoding vectors but not empty vector controls at the protein level (Figure 2) and in the RA cultures infected with lentivirus at the mRNA level (not shown).

We focused on *Hoxa5*, which is inappropriately repressed upon RA-mediated differentiation of Oct1 deficient ESCs. Infection of differentiating Oct1 deficient ESCs using constructs expressing Oct1 increased *Hoxa5* expression (Figure 2). These results suggest that complementing with Oct1, at least at this timepoint, restores at least some gene expression. We have added to the text, figure legends and Methods sections to incorporate these findings.

*Reviewer #3:*

*Issue 1: Figure 6. The authors compare RNA-seq data of parental vs. KO in both undifferentiated and differentiated conditions. The authors conclude that the KO show more gene expression differences during differentiation. However, it is important to show variability across parental and KO samples separately in both conditions. i.e. one would expect that cells are more heterogeneous during RA-differentiation compared to pluripotency state. So, it is expected to see more difference in RA-differentiation state.*

We appreciate the reviewer’s enthusiasm about the work. Although RA-differentiated cells are more heterogenous than undifferentiated ESCs, the variance in gene expression in the population of cells between RNAseq replicates is not. If it were, the *P*-values would not be robust and Figure 6 would not show differences on the Y-axis. To further solidify this, we are attaching a Pearson correlation of the RNAseq raw expression count data for all genes (Figure 9 and Figure 10). These figures recapitulates the finding that the differences in gene expression in differentiated vs. pluripotent are stronger than differences due to inducible knockout. Relevant here, replicate variance (as groups of nine squares) are all high at ~.99 regardless of the condition. We have modified the text to state that the findings in Figure 6 and associated figure supplements are not due to increased variance in the differentiated condition.

Author response image 1.**DOI:**
http://dx.doi.org/10.7554/eLife.20937.034

Author response image 2.**DOI:**
http://dx.doi.org/10.7554/eLife.20937.035

*Issue 2: Figure 7 (bottom). the known Oct4-Sox2 motif was enriched in 22.9% of Oct4 unique peaks as compared to Oct4 motif which was enriched 15.3%. this does not make sense as Oct4 motif is the exact first half of the Oct4-Sox2 composite motif. One would expect Oct4 motif to be present in all Oct4-Sox2 (22.9%) sites plus few more sites that only contain Oct4 motif. The authors need to clarify what those percentages mean in the legend.*

For identification of known motifs, both HOMER and CentriMo (MEME suite) assign a given sequence to the strongest weight matrices based in statistical power, which is stronger for longer motifs. We ran multiple simulations using synthetic and physiological sequences. With consensus Oct:Sox motifs the number of tested targets with recognized Oct4 and Oct4-*Sox2* motifs is the same. However, physiological transcription factor binding sites can deviate from consensus, especially in the case of longer sequences. Sequence mismatches lower the relative enrichment of the Oct4 motif as compared to Oct4-*Sox2*. Therefore, the presence of non-consensus but physiological sequences causes the composite motifs to score higher and, for a given threshold value, are reported by motif-finding software at a higher percentage as compared to an individual motif. We have added to the legend for this panel concisely explaining the percentages.